# RLeXplore: Accelerating Research in Intrinsically-Motivated Reinforcement Learning

**Mingqi Yuan**[*]                                         *mingqi.yuan@connect.polyu.hk*
*Department of Computing*
*The Hong Kong Polytechnic University*

**Roger Creus Castanyer**[*]                               *roger.creus-castanyer@mila.quebec*
*Mila Québec AI Institute & Université de Montréal*

**Bo Li**                                                  *comp-bo.li@polyu.edu.hk*
*Department of Computing*
*The Hong Kong Polytechnic University*

**Xin Jin**[†]                                             *jinxin@eitech.edu.cn*
*Ningbo Institute of Digital Twin*
*Eastern Institute of Technology, Ningbo*

**Wenjun Zeng**                                            *wzeng-vp@eitech.edu.cn*
*Ningbo Institute of Digital Twin*
*Eastern Institute of Technology, Ningbo*
*Fellow, IEEE and CAE*

**Glen Berseth**                                           *glen.berseth@mila.quebec*
*Mila Québec AI Institute & Université de Montréal*

**Reviewed on OpenReview:** *https://openreview.net/forum?id=B9BHjTN4z6*

## Abstract

Extrinsic rewards can effectively guide reinforcement learning (RL) agents in specific tasks. However, extrinsic rewards frequently fall short in complex environments due to the significant human effort needed for their design and annotation. This limitation underscores the necessity for intrinsic rewards, which offer auxiliary and dense signals and can enable agents to learn in an unsupervised manner. Although various intrinsic reward formulations have been proposed, their implementation and optimization details are insufficiently explored and lack standardization, thereby hindering research progress. To address this gap, we introduce RLeXplore, a unified, highly modularized, and plug-and-play framework offering reliable implementations of eight state-of-the-art intrinsic reward methods. Furthermore, we conduct an in-depth study that identifies critical implementation details and establishes well-justified standard practices in intrinsically-motivated RL. Our documentation, examples, and source code are available at `https://github.com/RLE-Foundation/RLeXplore`.

## 1 Introduction

Reinforcement learning (RL) provides a framework for training agents to solve tasks by learning from interactions with an environment. At the core of RL is the optimization of a reward function, where agents aim to maximize cumulative rewards over time (Sutton & Barto, 2018). However, in complex environments, defining extrinsic rewards that effectively guide an agent's learning process can be impractical, often requiring

---

[*]Equal contribution.
[†]Corresponding author.

domain-specific expertise. In practice, poorly defined extrinsic rewards can lead to sparse-reward settings, where RL agents struggle due to the lack of a meaningful learning signal (Burda et al., 2019a).

As the RL community tackles increasingly complex problems, such as training generally capable RL agents, there is a need for more autonomous agents capable of learning valuable behaviors without relying on dense supervision (Jiang et al., 2023). To address this challenge, the concept of intrinsic rewards has emerged as a promising approach in the RL community (Burda et al., 2019b; Pathak et al., 2017; Raileanu & Rocktäschel, 2020; Badia et al., 2020; Henaff et al., 2022; Pathak et al., 2019). Intrinsic rewards provide agents with additional learning signals, enabling them to explore and acquire skills across diverse environments beyond what extrinsic rewards alone can offer. However, computing intrinsic rewards often requires learning auxiliary models, heavy engineering, and performing expensive computations, making reproducibility challenging.

While several formulations of intrinsic rewards have been proposed (Pathak et al., 2017; Badia et al., 2020; Laskin et al., 2021), each with its potential benefits for improving agent learning, the field lacks a comprehensive understanding of the comparative advantages and challenges posed by these methods. Importantly, existing literature reports varying performance when using the same intrinsic rewards, reinforcing the need for a standardized framework and a deeper understanding of the optimization and implementation details.

In this paper, we introduce **RLeXplore**, an open-source library containing high-quality implementations of state-of-the-art (SOTA) intrinsic rewards. RLeXplore offers a plug-and-play framework for researchers working on intrinsically-motivated RL, enabling them to seamlessly integrate intrinsic rewards into their projects. Specifically, RLeXplore (1) facilitates fair comparisons across multiple intrinsic reward methods, (2) can be easily integrated with various RL frameworks, and (3) streamlines the development of new intrinsic reward methods. In Table 1, we compare the performance of the implementations in RLeXplore with the original results reported in previous works. In Appendix D, we provide the full details on reproducibility with RLeXplore.

Table 1: Summary of comparative results from our RLeXplore implementations and prior work. Right two columns report success rates (% of episodes solving the task), except for Procgen where we report mean episode rewards after training. See Appendix D for reproducibility and evaluation details.

| Environment | Intrinsic Reward | Original | RLeXplore |
|---|---|---|---|
| SuperMarioBros (10M Steps) | RIDE | 23% | **50**% |
| SuperMarioBros (10M Steps) | ICM | 30% | 30% |
| MiniGrid-DoorKey-16×16 (10M Steps) | ICM | 0% | **60**% |
| MiniGrid-DoorKey-16×16 (10M Steps) | RND | 0% | **60**% |
| MiniGrid-DoorKey-16×16 (10M Steps) | RIDE | **25**% | 12% |
| MiniGrid-DoorKey-8×8 (1M Steps) | RE3 | 50% | **95**% |
| MiniGrid-DoorKey-8×8 (1M Steps) | RND | 0% | **82**% |
| MiniGrid-DoorKey-8×8 (1M Steps) | ICM | 20% | **83**% |
| Procgen - 200 Mazes (25M Steps) | E3B | 3.00 | **4.10** |
| Procgen - 200 Mazes (25M Steps) | ICM | 2.50 | **5.90** |
| Procgen - 200 Mazes (25M Steps) | RND | 1.70 | **5.00** |

To support these capabilities, we have provided extensive documentation that includes detailed guides on using RLeXplore, along with comprehensive code tutorials. These resources are designed to make it straightforward for users to get started with RLeXplore, regardless of their prior experience with intrinsic rewards in RL. In Appendix D.6, we provide an overview of the main differences and advantages of RLeXplore compared to existing RL libraries.

We aim for the community to adopt RLeXplore as a standard tool for evaluating intrinsic reward methods, reducing implementation efforts, and mitigating inconsistencies in results and conclusions.

Our work presents a systematic study aimed at addressing gaps in understanding the critical implementation and optimization details of intrinsic rewards. We investigate the design of different intrinsic reward methods and (1) highlight challenges in the reproducibility of prior work, and (2) share highly performant

reimplementations of many popular methods. To guide our investigation, we formulate numerous questions, aiming to uncover the intricacies of intrinsic rewards and their impact on RL agent performance. Our results highlight the importance of thoughtful implementation design for intrinsic rewards, showing that naive implementations can lead to suboptimal performance. Through carefully studied design decisions, we demonstrate significant performance gains.

Our results show that with RLeXplore, RL agents can learn emergent behaviours autonomously, solving multiple levels of SuperMarioBros without task rewards. Additionally, we show that intrinsic rewards enable RL agents to obtain great performance on complex sparse-reward tasks like Procgen-Maze, MiniGrid, the ALE-5 hard-exploration tasks and Ant-UMaze.

## 2 Related Work

While some works have benchmarked intrinsic rewards in specific environments (Taiga et al., 2021; Wang et al., 2022; Laskin et al., 2021), their lack of detailed discussions on implementation and optimization leads to reproducibility problems (Voelcker et al., 2024). In this work, we introduce RLeXplore, a comprehensive framework that incorporates the most widely-used intrinsic rewards, which provides a standardized approach to enhance reproducibility, accelerate research, and facilitate the comparison of baselines in intrinsically-motivated RL. In the following, we overview existing formulations for intrinsic rewards of different natures and introduce the methods included in RLeXplore.

### 2.1 Count-Based Exploration

Count-based exploration methods provide intrinsic rewards by measuring the novelty of states, defined to be inversely proportional to the state visitation counts (Strehl & Littman, 2008; Tang et al., 2017; Machado et al., 2020; Jo et al., 2022). In finite state spaces, count-based methods perform near optimally (Strehl & Littman, 2008). For this reason, these methods have been established as appealing techniques for driving structured exploration in RL. However, they do not scale well to high-dimensional state spaces (Bellemare et al., 2016; Lobel et al., 2023). Pseudo-counts provide a framework to generalize count-based methods to high-dimensional and partially observed environments (Bellemare et al., 2016; Ostrovski et al., 2017; Martin et al., 2017). Burda et al. (2019b) proposed random network distillation (RND), which uses the prediction error against a fixed network as a learning signal that is correlated to counts. Recently, Henaff et al. (2022) proposed E3B and showed that the intrinsic objective provides a generalization of counts to high-dimensional spaces. In RLeXplore, we include Pseudo-counts, RND, and E3B as representatives of the state-of-the-art count-based methods.

### 2.2 Curiosity-Driven Exploration

Curiosity-based objectives train agents to interact with the environment seeking to experience outcomes that are not aligned with the agents' predictions (Aubret et al., 2023). Hence, curiosity-driven exploration usually involves training an agent to increase its knowledge about the environment (e.g., environment dynamics) (Stadie et al., 2015; Pathak et al., 2017; Yu et al., 2020). The intrinsic curiosity module (ICM) (Pathak et al., 2017; Burda et al., 2019a) learns a joint embedding space with inverse and forward dynamics losses and was the first curiosity-based method successfully applied to deep RL settings. Disagreement (Pathak et al., 2019) further extended ICM by using the variance over an ensemble of forward-dynamics models to compute curiosity. However, curiosity-driven methods are consistently found to be unsuccessful when the environment has irreducible noise (Savinov et al., 2019). To address the problem, Raileanu & Rocktäschel (2020) proposed RIDE, which uses the difference between two consecutive state embeddings as the intrinsic reward and encourages the agent to choose actions that result in significant state changes. In general, curiosity-based objectives remain amongst the most popular intrinsic rewards in deep RL applications to this day. In RLeXplore, we include ICM, Disagreement, and RIDE as representatives of the state-of-the-art curiosity-driven methods.

### 2.3 Global and Episodic Exploration

Towards more general and adaptive agents, recent works have studied decision-making problems in contextual Markov decision processes (MDPs) (e.g., procedurally-generated environments) (Raileanu & Rocktäschel, 2020; Henaff et al., 2022; Matthews et al., 2024). Contextual MDPs require episodic-level exploration, where novelty estimates are reset at the beginning of each episode. Henaff et al. (2023) showed that both global and episodic exploration modalities have unique benefits and proposed combined objectives that achieve remarkable performance across many MDPs of different structures. NGU (Badia et al., 2020) and RIDE (Raileanu & Rocktäschel, 2020) also instantiate both global and episodic bonuses. Inspired by this recent line of work, in this paper, we study novel combinations of objectives for exploration that achieve impressive results in contextual MDPs.

## 3 Background

We frame the RL problem considering a MDP (Bellman, 1957; Kaelbling et al., 1998) defined by a tuple $\mathcal{M} = (\mathcal{S}, \mathcal{A}, R, P, d_0, \gamma)$, where $\mathcal{S}$ is the state space, $\mathcal{A}$ is the action space, and $R : \mathcal{S} \times \mathcal{A} \to \mathbb{R}$ is the reward function, $P : \mathcal{S} \times \mathcal{A} \to \Delta(\mathcal{S})$ is the transition function that defines a probability distribution over $\mathcal{S}$, $d_0 \in \Delta(\mathcal{S})$ is the distribution of the initial observation $\boldsymbol{s}_0$, and $\gamma \in [0, 1)$ is a discount factor. The goal of RL is to learn a policy $\pi_{\boldsymbol{\theta}}(\boldsymbol{a}|\boldsymbol{s})$ to maximize the expected discounted return:

$$J_\pi(\boldsymbol{\theta}) = \mathbb{E}_\pi \left[ \sum_{t=0}^{\infty} \gamma^t R_t \right]. \tag{1}$$

Following prior work in intrinsically-motivated RL (Burda et al., 2019b), we consider an augmented reward function that combines both extrinsic and intrinsic components. The overall reward signal at time step $t$ is defined as:

$$R_t^{\text{total}} = R_t + \beta_t \cdot I_t, \tag{2}$$

where $I : \mathcal{S} \times \mathcal{A} \to \mathbb{R}$ is the intrinsic reward, $\beta_t = \beta_0 (1 - \kappa)^t$ is the exploration coefficient controlling the relative weight of intrinsic motivation over time, and $\kappa$ is a decay rate. This formulation enables the agent to balance task-specific objectives with exploratory behavior, particularly in sparse or deceptive reward settings. Finally, the augmented optimization objective is:

$$J_\pi(\boldsymbol{\theta}) = \mathbb{E}_\pi \left[ \sum_{t=0}^{\infty} \gamma^t (R_t + \beta_t \cdot I_t) \right]. \tag{3}$$

In Appendix A, we present a detailed overview of the SOTA intrinsic reward methods that we implement in RLeXplore.

## 4 RLeXplore

In this section, we present **RLeXplore**, a unified, highly-modularized and plug-and-play framework that currently provides high-quality and reliable implementations of eight SOTA intrinsic reward methods[1]. Comparing multiple intrinsic reward methods under fair conditions is challenging due to various confounding factors, such as using distinct RL frameworks (e.g., PPO (Schulman et al., 2017), DQN (Mnih et al., 2013), IMPALA (Espeholt et al., 2018)), optimization (e.g., reward and observation normalization, network architecture) and evaluation details (e.g., environment configuration, algorithm hyperparameters). RLeXplore is designed to provide a unified framework with standardized procedures for implementing, computing, and optimizing intrinsic rewards.

---

[1]RLeXplore complies with the MIT License.

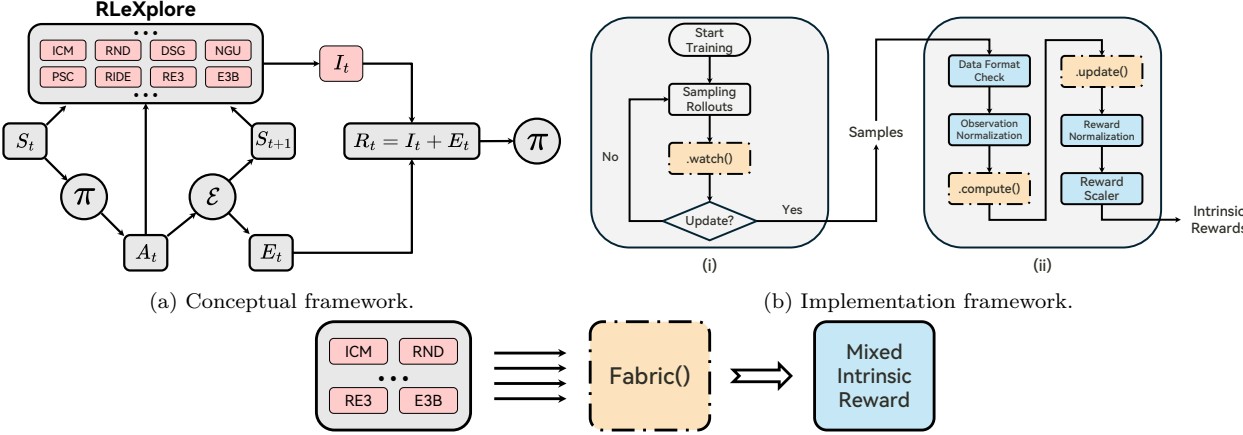

(a) Conceptual framework.

(b) Implementation framework.

(c) Create mixed intrinsic rewards using the `Fabric` class.

Figure 1: The workflow of RLeXplore. (a) RLeXplore provides a decoupled module for intrinsic rewards that integrates seamlessly with the RL training loop. RLeXplore implements 8 SOTA intrinsic rewards and adapts to the unmodified RL training loop. (b) RLeXplore monitors the agent-environment interactions and gathers data samples using the `.watch()` function. After collecting experience rollouts, RLeXplore computes the corresponding intrinsic rewards using the `.compute()` function and updates the auxiliary models via the `.update()` function. (c) RLeXplore provides a `Fabric` class that allows developers to combine multiple intrinsic rewards in an elegant manner. In Appendix C.6 we provide more details on how to add new intrinsic rewards to RLeXplore.

## 4.1 Architecture

The core design decision of RLeXplore involves decoupling the intrinsic reward modules from the RL optimization algorithms, which enables our intrinsic reward implementations to be integrated with any desired RL algorithm (or existing library, see Appendix C and the official integration examples). Figure 1 illustrates the basic workflow of RLeXplore, which consists of two parts: data collection (i.e., policy rollout) and reward computation.

Commonly, at each time step, the agent receives observations from the environment and predicts actions. The environment then executes the actions and returns feedback to the agent, which consists of a next observation, a reward, and a terminal signal. During the data collection process, the `.watch()` function is used to monitor the agent-environment interactions. For instance, E3B (Henaff et al., 2022) updates an estimate of an ellipsoid in an embedding space after observing every state. At the end of the data collection rollouts, `.compute()` computes the corresponding intrinsic rewards. Note that `.compute()` is only called once per rollout using batched operations, which makes RLeXplore a highly efficient framework. Additionally, RLeXplore provides several utilities for reward and observation normalization. Finally, the `.update()` function is called immediately after `.compute()` to train the reward module if necessary (e.g., train the forward dynamics models in Disagreement (Pathak et al., 2019) or the predictor network in RND (Burda et al., 2019b)). Appendix C illustrates the usage of the aforementioned functions. All operations are subject to the standard workflow of the Gymnasium API (Towers et al., 2023).

In particular, recent research (Henaff et al., 2023) has highlighted that mixed intrinsic rewards can significantly promote the agent's exploration capability by providing comprehensive exploration incentives. In RLeXplore, we provide a `Fabric` class that allows developers to combine multiple intrinsic rewards in an elegant manner, as illustrated in Appendix C.3.

RLeXplore offers several benefits to the research community:

- For researchers seeking reliable tools for benchmarking and general applications: RLeXplore provides high-quality implementations of popular intrinsic reward methods, useful in both research and practical applications. It can be seamlessly integrated with existing RL libraries. We provide specific examples of integrating RLeXplore with Stable-Baselines3 (Raffin et al., 2021), CleanRL (Huang et al., 2022b), and RLLTE (Yuan et al., 2023) in Appendix C, F, and G.

- For developers experimenting with new intrinsic rewards: RLeXplore offers modular components, such as various embedding networks and a standardized workflow. This setup facilitates the creation, modification, and testing of new ideas. Detailed examples are available in the code repository and documentation.

- For promoting collaboration and accelerating progress: We have published a space using Weights & Biases (W&B) to store reusable experiment results on recognized benchmarks. This initiative aims to enhance collaboration within the research community and speed up progress by providing easy access to established benchmark results.

### 4.2 Algorithmic Baselines

In RLeXplore, we implement eight widely-recognized intrinsic reward methods spanning the different categories described in Section 2, namely ICM (Pathak et al., 2017), RND (Burda et al., 2019b), Disagreement (Pathak et al., 2019), NGU (Badia et al., 2020), PseudoCounts (Badia et al., 2020), RIDE (Raileanu & Rocktäschel, 2020), RE3 (Seo et al., 2021), and E3B (Henaff et al., 2022), respectively. We selected them based on the following tenet:

- The algorithm represents a unique design philosophy;

- The algorithm achieved superior performance on well-recognized benchmarks;

- The algorithm can adapt to arbitrary tasks and can be combined with arbitrary RL algorithms.

For detailed descriptions of each method, we refer the reader to Appendix A.

## 5 Experiments

Our experiments aim to achieve two main objectives: (i) highlight how intrinsic reward methods are sensitive to implementation details, and (ii) identify the best algorithmic and design choices to ensure high performance across various sparse-reward environments to demonstrate the generality and robustness of our framework.

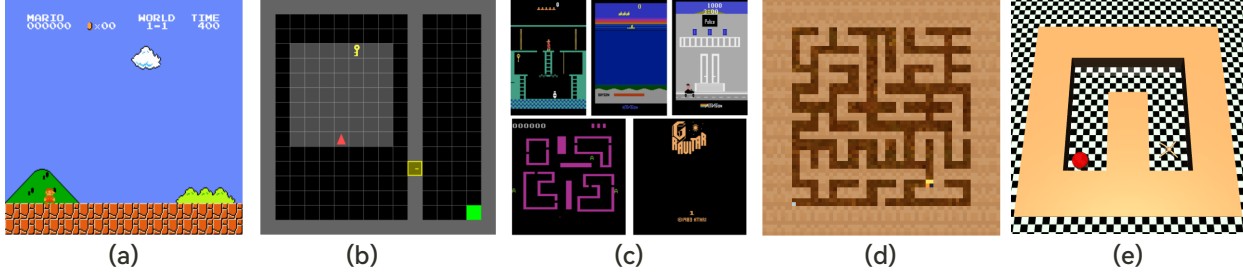

Figure 2: Screenshots of the selected exploration games. (a) *SuperMarioBros.* (b) *MiniGrid.* (c) *ALE-5.* (d) *Procgen-Maze.* (e) *Ant-UMaze.*

First, we use *SuperMarioBros (SMB)* without access to the environment's rewards to study the low-level implementation details of intrinsic reward methods that drive robust exploration. We selected *SMB* because effective exploration within this environment strongly correlates with task performance, making it an excellent benchmark for measuring the efficacy of exploration techniques. This environment has been widely used in previous studies on exploration in RL (Pathak et al., 2019; Raileanu & Rocktäschel, 2020; Burda et al., 2019a). To further generalize our findings, we also use the *MiniGrid-DoorKey-16×16 (MGD)* environment, which is challenging due to the sparse rewards, making it difficult to solve with classical RL algorithms[2]. The effectiveness of intrinsic rewards in *MiniGrid* environments has also been highlighted in prior works (Raileanu & Rocktäschel, 2020; Henaff et al., 2022; 2023). With these two environments we aim to study the implementation details in both reward-free and sparse-reward tasks.

---

[2]https://minigrid.farama.org/environments/minigrid/DoorKeyEnv

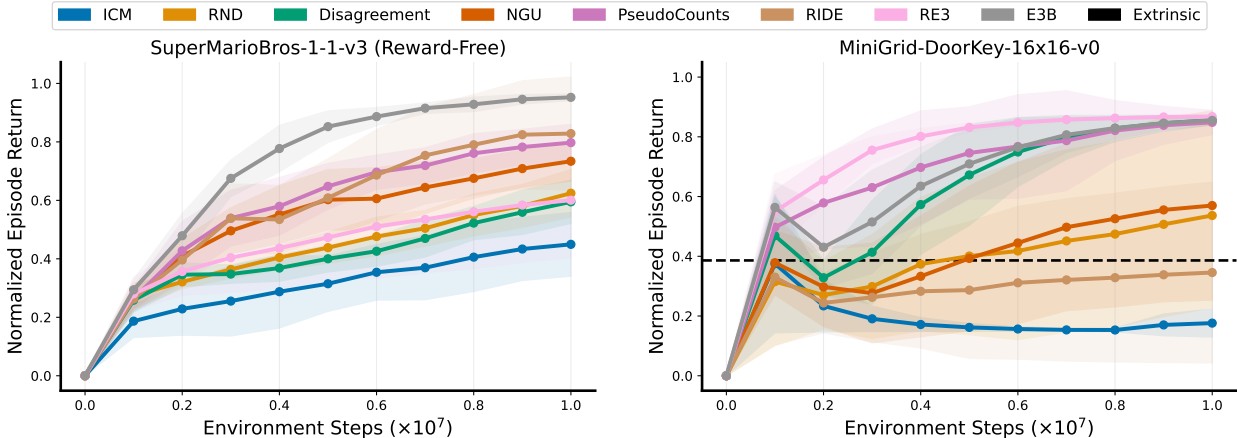

Figure 3: Episode returns achieved by the intrinsic rewards in RLeXplore. (left) *SuperMarioBros* without access to the task rewards. (right) *MiniGrid-DoorKey-16×16* with sparse rewards.

Secondly, to showcase the generalizability of RLeXplore, we evaluate our implementations in additional sparse-reward environments, including *Procgen*, *MiniGrid*, *Ant-UMaze*, and the set of five hard-exploration games in the *arcade learning environment (ALE)* suite. These experiments are designed to test how well our methods balance the use of dense intrinsic rewards with sparse extrinsic rewards across a variety of tasks. The complete set of learning curves for all the experiments is shown in Appendix E.

Lastly, we explore recent advancements in using combined intrinsic rewards (Henaff et al., 2023) to enhance exploration in contextual MDPs. Specifically, we use the full set of levels in *SMB* to evaluate how well both single and combined intrinsic rewards can explore various game versions and generalize their exploration across different levels.

In the following sections, we present results from *SMB* and *MiniGrid* for objective (i) and from *Procgen-Maze* for objective (ii). Additionally, in Appendix D, we show that using RLeXplore, we are able to reproduce and improve the performance reported in previous works for many intrinsic rewards and across multiple environments.

The design of these experiments is driven by our primary goal: to provide a general and reliable set of intrinsic reward implementations within a user-friendly framework. Instead of attempting to benchmark all methods across every possible domain, we focus on verifying the generality of each method within a carefully selected subset of popular exploration tasks. Finally, we provide the experimental details of test benchmarks and method configurations in Appendix B.

## 5.1 Low-level Implementation Details of Intrinsic Rewards

The performance of intrinsic rewards is affected by various factors that tend to vary with the complexity of the task. For instance, the RL algorithm used for optimization, the architecture of the networks, algorithm-specific hyperparameters, and the joint optimization of intrinsic and extrinsic rewards. As a result, implementing and reproducing intrinsic reward methods is challenging. To tackle this problem, we first formulate five questions to investigate how various low-level implementation details impact the training of intrinsically motivated agents. We first define an initial baseline configuration for optimizing the intrinsic rewards, shown in Table 2. These baseline settings are selected based on the most common configurations reported in the literature. Next, we address each question by modifying only one hyperparameter in the baseline configuration at a time. Finally, we evaluate the performance of these intrinsic rewards with the best parameters gathered in each question. All the experiments are con-

Table 2: Details of the baseline settings.

| Hyperparameter | Value |
| --- | --- |
| Observation norm. | RMS |
| Reward norm. | RMS |
| Weight init. | Orthogonal |
| Update proportion | 100% |
| with LSTM | False |

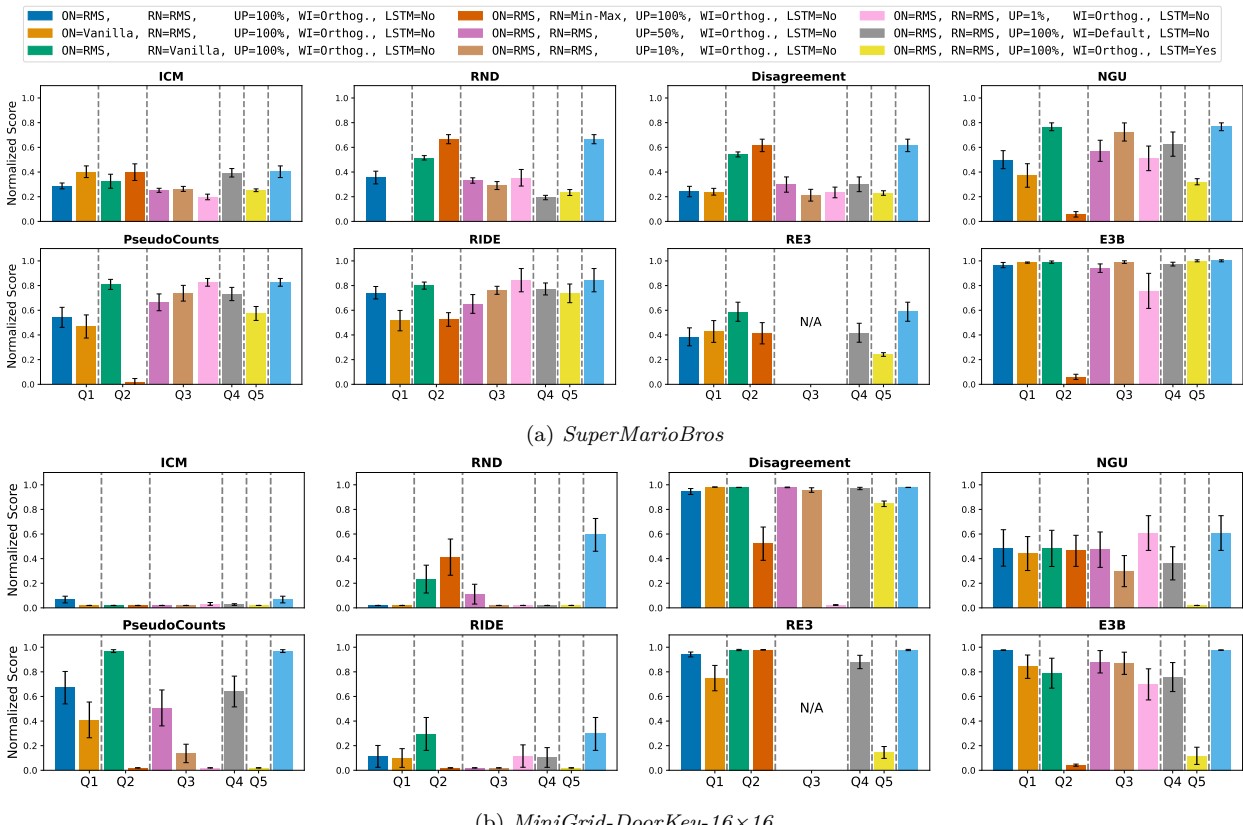

(a) *SuperMarioBros*

(b) *MiniGrid-DoorKey-16×16*

Figure 4: Results for Q1, Q2, Q3, Q4, and Q5 in *SMB* (top) and *MGD* (bottom), which are normalized by the maximum score possibly achieved in the task. Here, **ON** is the observation normalization, **RN** is the reward normalization, **UP** is the update proportion, and **WI** is the weight initialization. The bar with a hatch mark is **Combined**, which refers to the results of using the best hyperparameters gathered in each question. Since RE3 only employs a fixed, randomly initialized neural network for encoding observations, there are no values in Q3. All the results are aggregated over 10 seeds, and each run uses 10M environment interactions.

ducted using *SMB* and *MGD* to investigate the effects in sparse-rewards and reward-free (i.e., without access to extrinsic rewards) scenarios, respectively.

Importantly, as shown in Figure 1, we keep the PPO hyperparameters fixed and the overall RL training loop unmodified throughout all the experiments in the paper in order to isolate the effect of the questions on the intrinsic reward components. Previous work has shown that PPO has many implementation details that are key to achieving great performance (Huang et al., 2022a; Engstrom et al., 2020). In the following, we study implementation details for the intrinsic reward components. The fixed PPO hyperparameters are shown in Table 8.

---

**Q1: The impact of observation normalization.**

---

Observation normalization is crucial in deep learning to avoid numerical instabilities during optimization. Image observations, where each pixel value typically ranges from 0 to 255 per colour channel, are commonly normalized to a range of 0 to 1 using Min-Max normalization by dividing each pixel value by 255. However, previous studies suggest that Min-Max normalization may not be ideal for all representation learning algorithms (Burda et al., 2019b).

In Q1, we compare Min-Max normalization with using an exponential moving average (EMA) of the mean and standard deviation for observation normalization (RMS) for the inputs to the intrinsic reward modules. RMS normalizes observations by subtracting the running mean and dividing it by the running standard

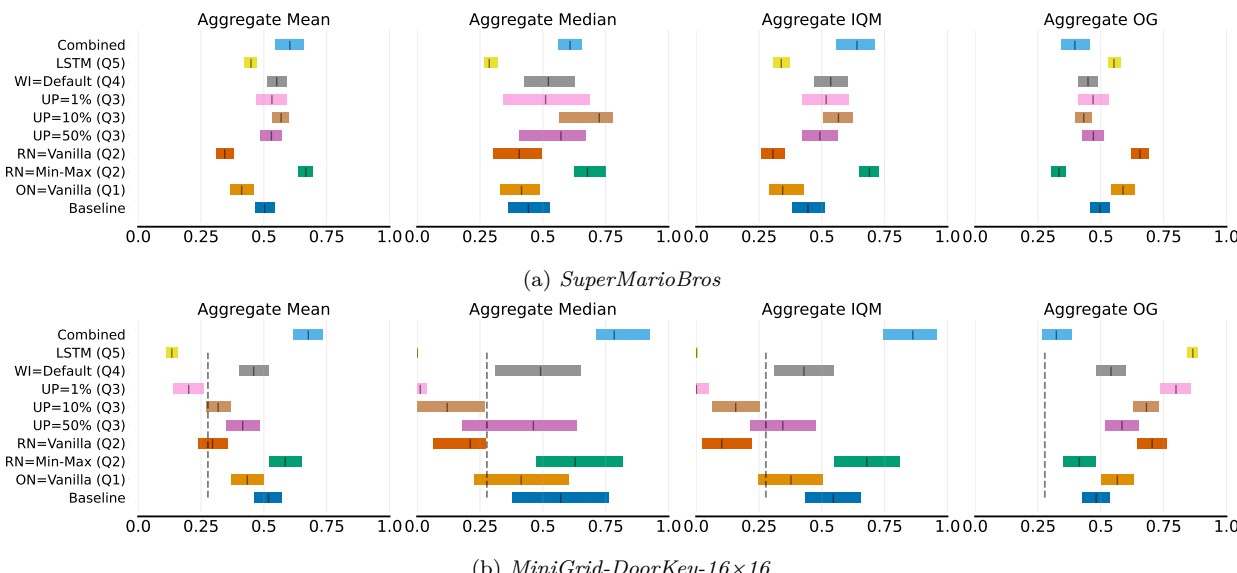

Figure 5: Aggregated performance of the eight intrinsic rewards with different low-level hyperparameters over 10 random seeds. The vertical dashed line represents the performance of the extrinsic agent, which only has access to the task rewards. Here, **ON** is the observation normalization, **RN** is the reward normalization, **UP** is the update proportion, **WI** is the weight initialization, **IQM** is the interquartile mean, **OG** is the optimality gap (lower is better), and **Combined** refers to the results of using the best hyperparameters gathered in each question. All the computation is performed using the Rliable (Agarwal et al., 2021) library.

deviation of all observations collected by the agent thus far. Our results shown in Figures 4 and 5 indicate that using RMS for observation normalization generally reduces the variance and achieves better asymptotic performance across all the environments of study. Importantly, some intrinsic rewards, such as RND, NGU, PseudoCounts, and RIDE, benefit significantly from RMS normalization. Critically, RND achieves zero rewards in SMB if observations are not normalized with RMS. These results indicate that RMS normalization is important for intrinsic reward methods that use random networks, since the lack of normalization can result in the embeddings produced by the random networks carrying very little information about the inputs (Burda et al., 2019b).

> **Q2: The impact of reward normalization.**

Similarly to Q1, reward normalization can have a large impact when using deep neural networks to compute the intrinsic rewards, since the scale of these rewards can be arbitrary and vary significantly over time. To mitigate the non-stationarity of intrinsic rewards, in Q2, we compare three normalization approaches for the reward outputs of the intrinsic reward modules: (1) Min-Max normalization, (2) using an RMS of the standard deviation, and (3) no reward normalization.

Reward normalization smooths the optimization process, which can be beneficial for stability but can lead to slower convergence (Burda et al., 2019b). Our findings show that almost all intrinsic rewards critically require some form of reward normalization, as agents fail to explore without normalized rewards. Importantly, the latter applies to all the environments that we experiment with. Additionally, while RMS is generally the default strategy for reward normalization, our results in Figure 5 show that Min-Max normalization is a more robust option in *SMB*, improving the performance and reducing the variance of the majority of the methods.

> **Q3: The co-learning dynamics of policies and auxiliary tasks for intrinsic rewards.**

Optimizing intrinsic rewards in deep RL often involves training additional networks for auxiliary tasks (e.g., predictor network in RND, inverse dynamics encoder in ICM, forward dynamics encoders in Disagreement). However, managing the co-learning dynamics of the auxiliary networks and policies is challenging. In Q3, we explore three update strategies for the auxiliary networks in the intrinsic reward modules: (1) updating them at the same frequency as the policy, (2) updating them 50% of the time, (3) updating them 10% of the time, and (4) updating them 1% of the time. This comparison sheds light on the trade-off between the number of gradient updates in the auxiliary networks and the performance of the policy. Additionally, lower update frequencies have the benefit of reducing computational overhead and training time by limiting the number of gradient updates required.

Our findings indicate that the auxiliary networks generally perform robustly across the range of studied update frequencies. Additionally, there is no clear configuration that seems generally better for all intrinsic rewards across environments, rendering this implementation detail worth tuning for specific environments.

---

### Q4: The impact of weight initialization.

---

Weight initialization plays a crucial role in optimizing deep neural networks, enabling faster convergence. In Q4, we compare two approaches for weight initialization in the auxiliary networks of the intrinsic reward modules: (1) orthogonal weight initialization and (2) uniform weight initialization (PyTorch's default). Note that again, the policy and value networks remain unchanged.

Our results highlight the importance of weight initialization in intrinsically-motivated RL. Specifically, we found that orthogonal weight initialization is beneficial for most intrinsic rewards, regardless of their specific optimization tasks (e.g., inverse dynamics, forward dynamics), and even in random networks (e.g., RND and RE3). This benefit is evidenced by reduced variance in episode returns and generally higher mean returns. This observation aligns with previous research indicating that orthogonal weight initialization can improve performance stability in deep RL agents (Huang et al., 2022a; Engstrom et al., 2020). Importantly, RND is the intrinsic reward method that shows the highest variability for this implementation detail, where orthogonal weight initialization works better in *SMB* but worse than uniform initialization in *MGD*.

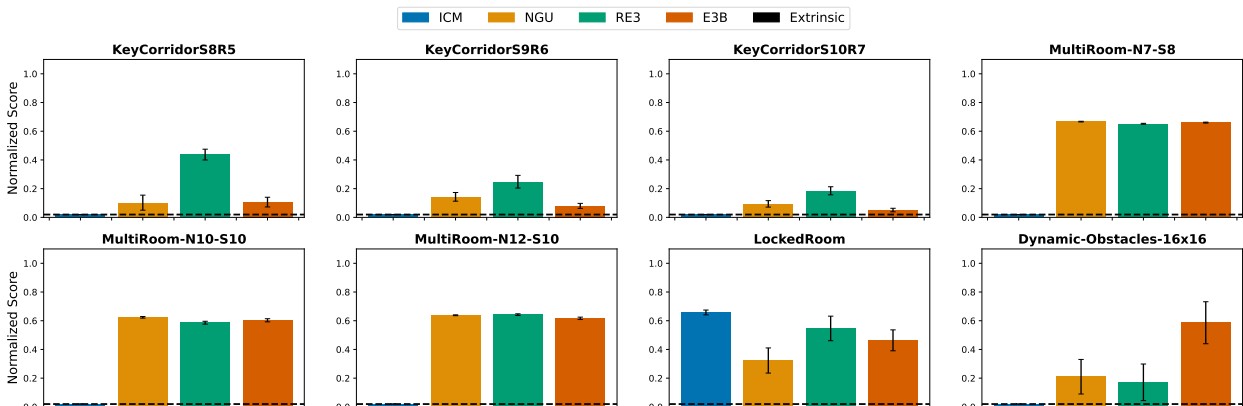

Figure 6: Performance of four selected intrinsic rewards in RLeXplore on the top eight most challenging tasks of the MGD suite. The solid line and shaded regions represent the mean and standard deviation computed with five random seeds, respectively.

---

### Q5: Is memory required to optimize intrinsic rewards?

---

In Q5, we investigate whether the intrinsic rewards included in RLeXplore benefit from memory-enabled architectures. We compare the optimization of intrinsic rewards using a vanilla policy network and one equipped with a long-short-term memory (LSTM) (Hochreiter & Schmidhuber, 1997) module while keeping PPO as the RL backbone algorithm.

Some intrinsic reward methods exhibit significantly lower performance when using LSTM policies. This observation aligns with the fact that LSTMs provide episodic context to policies, whereas most intrinsic reward methods define exploration as a global problem.

Finally, we use the best-performing implementation details observed from Q1-5 to experiment in the set of most challenging exploration tasks from *MiniGrid*. Our results in Figure 6 show that with our implementations of intrinsic rewards in RLeXplore, researchers can make progress in training RL agents in challenging tasks where vanilla RL agents are unable to learn due to the sparsity of the task rewards. In summary, by systematically addressing the implementation details, our work significantly enhances the reproducibility of intrinsic reward methods. These thoughtful design choices not only improve performance but also ensure that our implementations can be reliably reproduced and generalized across various environments.

## 5.2 Combination of Intrinsic and Extrinsic Rewards

> **Q6: Joint optimization of intrinsic and extrinsic rewards.**

In sparse-reward environments, the objective is for agents to explore the state space by optimizing intrinsic rewards until they discover the task rewards, at which point they should focus solely on optimizing the task rewards. However, many intrinsically motivated RL applications naively optimize the sum of intrinsic and extrinsic rewards, potentially leading to learning fuzzy value functions and suboptimal policies (Castanyer et al., 2023). In this section, we compare this common approach with learning two separate value functions, one for each reward function (Burda et al., 2019b). The advantages of the latter include the ability to disentangle the effects of intrinsic and extrinsic rewards on the agent's behaviour, leading to cleaner learning dynamics and potentially more efficient exploration. In these settings, both value functions are used during the advantage estimation phase of PPO. Specifically, we compute separate GAE values - one using the intrinsic value function and one using the extrinsic value function. The resulting advantages are then summed to compute the policy loss term for PPO. This separation facilitates more accurate advantage estimates for each reward type, leading to improved learning dynamics.

For this analysis, we used the *Procgen-Maze* task (Cobbe et al., 2020) as a sparse-reward benchmark. RL agents often struggle to learn meaningful behaviours from the extrinsic reward alone in this task. We evaluate different variants of the task (e.g., 1 maze vs. 200 mazes) to examine singleton versus contextual MDPs. We note that in our framework, we do not provide different context information to the agents for singleton versus contextual MDPs (e.g., the context ID). We refer to these frameworks to formalize the agent-environment interaction when the environment remains static throughout training (i.e., singleton - 1 maze) versus when it varies at each episode (i.e., contextual - a different maze at each episode).

Figure 7 demonstrates that learning two separate value functions (Huang et al., 2022b), which we refer to as the *TwoHead* architecture, outperforms the naive approach of simply adding the two rewards in the complex sparse-reward environment of *Procgen-Maze*, both in singleton and contextual settings. Importantly, all methods outperform the extrinsic agent, especially in the *1 Maze* environment.

## 5.3 Unlocking the Potential of Intrinsic Rewards

Q1-6 extensively discuss the tuning of intrinsic rewards under both normal and reward-free scenarios, revealing significant insights into the optimization processes. However, we aim to delve deeper into the capabilities of intrinsic rewards to address the evolving challenges in the RL community. Specifically, in Q7, we investigate recent developments in the exploration literature in RL, such as combined intrinsic rewards and exploration in contextual MDPs. For our experiments, we use the *SMB-RandomStages* environment variant, where agents play a different level in the game at each episode. Our results indicate that the recent developments in combined intrinsic rewards merit further research, as we demonstrate that such methods can enable agents to learn exploratory behaviours of exceptional quality in both singleton and contextual MDPs.

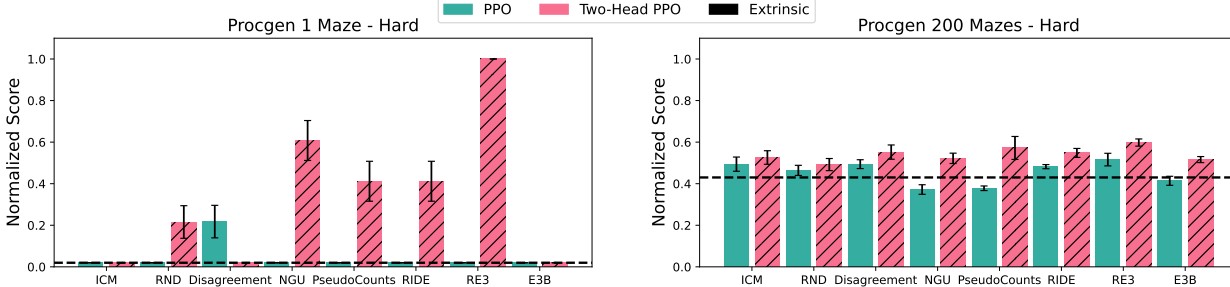

Figure 7: (Left) During training, the extrinsic agent struggles to find the goal in the selected Maze, resulting in a reward of 0. While some intrinsic reward methods yield occasional non-zero rewards, the methods perform significantly better when intrinsic and extrinsic value estimation are decoupled using two distinct value heads in the agent's network. (Right) In the Procgen variant, where each maze represents a unique level, the baseline extrinsic agent achieves the goal 50% of the time, and intrinsic rewards don't outperform the baseline significantly. We note that the presence of easier levels, where the goal may occasionally be near the agent's starting point results in generally less sparse rewards and an easier task to learn.

> **Q7: The performance of mixed intrinsic rewards.**

We run experiments using all the levels in the game of *SMB*, and we sample them uniformly during training. As in Q1-5, we do not use the extrinsic reward for training the agents but use it as an evaluation metric to show how much agents actively explore the environment.

Our results show that combined objectives enable emergent behaviours of much better quality than single objectives. Interestingly, E3B and RIDE are the best performing single objectives, and E3B+RIDE also achieves the highest performance among all the combinations. Similarly, RND and ICM, combined with other intrinsic rewards, outperform their original performance. This indicates that different intrinsic rewards can provide orthogonal gains that can be leveraged together.

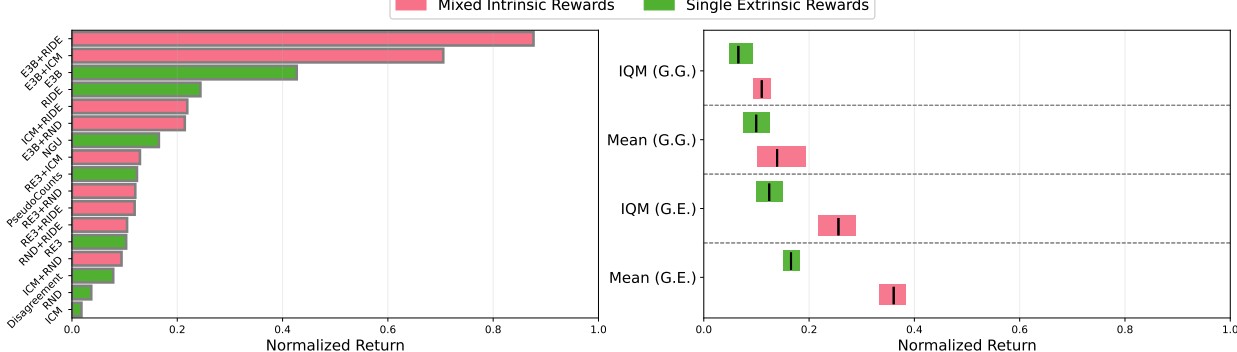

Figure 8: (Left) The performance ranking of single and mixed intrinsic rewards on the *SuperMarioBrosRandomLevels*. As expected, episodic bonuses (such as E3B and RIDE) demonstrate superior performance, attributed to the environment's non-singleton MDP nature. (Right) Overall performance comparisons between the single and mixed intrinsic rewards. Here, **G.E.** denotes the six "global+episodic" combinations, and **G.G.** denotes the three "global+global" combinations, as illustrated in Table 5.

# 6 Conclusion

Our work introduces RLeXplore, a comprehensive open-source repository that not only implements state-of-the-art intrinsic rewards but also provides a systematic evaluation framework for understanding their impact on agent performance. Our results show that with RLeXplore, RL agents can learn emergent behaviours autonomously, solving multiple levels of *SuperMarioBros* without task rewards. Additionally, we show that

intrinsic rewards enable RL agents to obtain great performance on complex sparse-reward tasks like *Procgen-Maze*, *MiniGrid*, the *ALE-5 hard-exploration tasks* and *Ant-UMaze*. Finally, RLeXplore facilitates further research in mixed intrinsic rewards (Henaff et al., 2023), uncovering the potential of such methods.

Through our study, we emphasize the importance of thoughtful implementation design, demonstrating that well-considered approaches lead to significant performance gains over naive implementations. Our contributions extend to establishing standardized practices for implementing and optimizing intrinsic rewards, laying the groundwork for future advancements in intrinsically motivated RL.

RLeXplore is designed to benchmark end-to-end intrinsic reward methods. These end-to-end methods are more commonly used and under-evaluated by the community. Skill-based algorithms, which typically involve separate phases for skill discovery and skill learning, are more complex and left for future work. For an alternative perspective that includes skill-based approaches, we refer readers to the unsupervised RL benchmark by Laskin et al. (2021). Additionally, RLeXplore was designed with accessibility in mind, ensuring that the implemented methods can be run on standard computational resources by any researcher. To maintain this accessibility, we have not included more complex and potentially powerful methods like BYOL-Explore (Guo et al., 2022) or RECODE (Kapturowski et al.). These methods are not open-source and have been optimized exclusively with non-open-source RL algorithms, which further limits their integration into RLeXplore.

## Acknowledgments

This work is funded, in part, by HKSAR RGC under Grant No. PolyU 15224823, the Guangdong Basic and Applied Basic Research Foundation under Grant No. 2024A1515011524, the NSFC under Grant No. 62302246, the ZJNSFC under Grant No. LQ23F010008, and the Ningbo under Grants No. 2023Z237 & 2024Z284 & 2024Z289 & 2023CX050011 & 2025Z038. We thank the high-performance computing center at Eastern Institute of Technology and Ningbo Institute of Digital Twin for providing the computing resources. We also want to acknowledge funding support from NSERC and CIFAR, and compute support from Digital Research Alliance of Canada, Mila IDT and Nvidia.

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

# A   Algorithmic Baselines

**ICM** (Pathak et al., 2017). ICM leverages an inverse-forward model to learn the dynamics of the environment and uses the prediction error as the curiosity reward. Specifically, the inverse model inferences the current action $\boldsymbol{a}_t$ based on the encoded states $\boldsymbol{e}_t$ and $\boldsymbol{e}_{t+1}$, where $\boldsymbol{e} = \psi(\boldsymbol{s})$ and $\psi(\cdot)$ is an embedding network. Meanwhile, the forward model $f$ predicts the encoded next-state $\boldsymbol{e}_t$ based on $(\boldsymbol{e}_t, \boldsymbol{a}_t)$. Finally, the intrinsic reward is defined as

$$I_t = \|f(\boldsymbol{e}_t, \boldsymbol{a}_t) - \boldsymbol{e}_{t+1}\|_2^2. \tag{4}$$

**RND** (Burda et al., 2019b). RND produces intrinsic rewards via a self-supervised manner, in which a predictor network $\hat{f}$ is trained to approximate a fixed and randomly-initialized target network $\hat{f}$. As a result, the agent is motivated to explore unseen parts of the state space. The intrinsic reward is defined as

$$I_t = \|\hat{f}(\boldsymbol{s}_{t+1}) - f(\boldsymbol{s}_{t+1})\|_2^2. \tag{5}$$

**Disagreement** (Pathak et al., 2019). Disagreement is a variant of ICM that leverages an ensemble of forward models and calculates the intrinsic reward as the variance among these models. Accordingly, the intrinsic reward is defined as

$$I_t = \mathrm{Var}\{f_i(\boldsymbol{e}_t, \boldsymbol{a}_t)\}, i = 0, \ldots, N. \tag{6}$$

**NGU** (Badia et al., 2020). NGU is a mixed intrinsic reward approach that combines global and episodic exploration and the first method to achieve non-zero rewards in the game of *Pitfall!* without using demonstrations or hand-crafted features. The intrinsic reward is defined as

$$I_t = \min\{\max\{\alpha_t\}, C\}/\sqrt{N_{\mathrm{ep}}(\boldsymbol{s}_t)}, \tag{7}$$

where $\alpha_t$ is a life-long curiosity factor computed following the RND method, $C$ is a chosen maximum reward scaling, and $N_{\mathrm{ep}}$ is the episodic state visitation frequency computed by pseudo-counts.

**PseudoCounts** (Badia et al., 2020). Pseudo-counts has been widely used in count-based exploration approaches (Bellemare et al., 2016; Ostrovski et al., 2017) with diverse implementations like neural density models. In this paper, we follow NGU (Badia et al., 2020) that computes pseudo-counts via $k$-nearest neighbor estimation, which is highly efficient and can be applied to arbitrary tasks. Given the encoded observations $\{\boldsymbol{e}_0, \ldots, \boldsymbol{e}_{T-1}\}$ visited in the an episode, we have

$$\sqrt{N_{\mathrm{ep}}(\boldsymbol{s}_t)} \approx \sqrt{\sum_{\tilde{\boldsymbol{e}}_i} K(\tilde{\boldsymbol{e}}_i, \boldsymbol{e}_t)} + c, \tag{8}$$

where $\tilde{\boldsymbol{e}}_i$ is the first $k$ nearest neighbors of $\boldsymbol{e}$, $K$ is a Dirac delta function, and $c$ guarantees a minimum amount of pseudo-counts. Finally, the intrinsic reward is defined as

$$I_t = 1/\sqrt{N_{\mathrm{ep}}(\boldsymbol{s}_t)} \tag{9}$$

**RIDE** (Raileanu & Rocktäschel, 2020). RIDE is designed based on ICM that learns the dynamics of the environment and rewards significant state changes. Accordingly, the intrinsic reward is defined as

$$I_t = \|\boldsymbol{e}_{t+1} - \boldsymbol{e}_t\|_2/\sqrt{N_{\mathrm{ep}}(\boldsymbol{s}_{t+1})}, \tag{10}$$

where $N_{\mathrm{ep}}(\boldsymbol{s}_{t+1})$ is used to discount the intrinsic reward and prevent the agent from lingering in a sequence of states with a large difference in their embeddings.

**RE3** (Seo et al., 2021). RE3 is an information theory-based and computation-efficient exploration approach that aims to maximize the Shannon entropy of the state visiting distribution. In particular, RE3 leverages a random and fixed neural network to encode the state space and employs a $k$-nearest neighbor estimator

to estimate the entropy efficiently. Then, the estimated entropy is transformed into particle-based intrinsic rewards. Specifically, the intrinsic reward is defined as

$$I_t = \frac{1}{k} \sum_{i=1}^{k} \log(\|\boldsymbol{e}_t - \tilde{\boldsymbol{e}}_t^i\|_2 + 1). \tag{11}$$

**E3B** (Henaff et al., 2022). E3B provides a generalization of count-based rewards to continuous spaces. E3B learns a representation mapping from observations to a latent space (e.g., using inverse dynamics). At each episode, the sequence of latent observations parameterizes an ellipsoid (Li et al., 2010; Auer, 2002; Dani et al., 2008), which is used to measure the novelty of the subsequent observations. In tabular settings, the E3B ellipsoid reduces to the table of inverse state-visitation frequencies (Henaff et al., 2022). Given a feature encoding $f$, at each time step $t$ of the episode the elliptical bonus $I_t$ is defined as follows:

$$I_t = f(\boldsymbol{s}_t)^T C_{t-1} f(\boldsymbol{s}_t), \tag{12}$$

$$C_{t-1} = \sum_{i=1}^{t-1} f(\boldsymbol{s}_i) f(\boldsymbol{s}_i)^T + \lambda \mathbf{I}, \tag{13}$$

where $f$ is the learned representation mapping, $C_{t-1}$ is the episodic ellipsoid (Henaff et al., 2022), $\lambda$ is a scalar coefficient, and $\mathbf{I}$ is the identity matrix.

## B   Experimental Settings

### B.1   Benchmark Selection

We evaluate the RLeXplore framework on multiple recognized benchmarks, which are specifically designed to evaluate the exploration capability of RL agents. We select SuperMarioBros (Kauten, 2018), MiniGrid (Chevalier-Boisvert et al., 2023), Procgen (Cobbe et al., 2020), Arcade learning environment (ALE) (Bellemare et al., 2013), and Gymnasium-Robotics (de Lazcano et al., 2024) for our experiments, which sufficiently spans the existing RL benchmarks. Table 3 provides the details of these selected environments, including their observation, action, and reward spaces.

Table 3: Details of the environments used in our experiments.

| Benchmark | Environment | Observation Space | Action Space | Reward Space |
|---|---|---|---|---|
| SuperMarioBros | World1-Stage1-v3 | Box(0, 255, (240, 256, 3)) | Discrete(7) | N/A |
| SuperMarioBros | RandomStages-v3 | Box(0, 255, (240, 256, 3)) | Discrete(7) | N/A |
| MiniGrid | DoorKey-16×16-v0 | Box(0, 255, (3, 7, 7)) | Discrete(7) | Sparse |
| MiniGrid | DoorKey-8×8 | Box(0, 255, (3, 7, 7)) | Discrete(7) | Sparse |
| MiniGrid | KeyCorridorS8R5-v0 | Box(0, 255, (3, 7, 7)) | Discrete(7) | Sparse |
| MiniGrid | KeyCorridorS9R6-v0 | Box(0, 255, (3, 7, 7)) | Discrete(7) | Sparse |
| MiniGrid | KeyCorridorS10R7-v0 | Box(0, 255, (3, 7, 7)) | Discrete(7) | Sparse |
| MiniGrid | MultiRoom-N7-S8-v0 | Box(0, 255, (3, 7, 7)) | Discrete(7) | Sparse |
| MiniGrid | MultiRoom-N10-S10-v0 | Box(0, 255, (3, 7, 7)) | Discrete(7) | Sparse |
| MiniGrid | MultiRoom-N12-S10-v0 | Box(0, 255, (3, 7, 7)) | Discrete(7) | Sparse |
| MiniGrid | LockedRoom-v0 | Box(0, 255, (3, 7, 7)) | Discrete(7) | Sparse |
| MiniGrid | Dynamic-Obstacles-16x16-v0 | Box(0, 255, (3, 7, 7)) | Discrete(3) | Sparse |
| Procgen | Maze, num_levels=200 | Box(0, 255, (3, 64, 64)) | Discrete(15) | Sparse |
| Procgen | Maze, num_levels=0 | Box(0, 255, (3, 64, 64)) | Discrete(15) | Sparse |
| ALE | Gravitar | Box(0, 255, (4, 84, 84)) | Discrete(18) | Dense |
| ALE | MontezumaRevenge | Box(0, 255, (4, 84, 84)) | Discrete(18) | Dense |
| ALE | PrivateEye | Box(0, 255, (4, 84, 84)) | Discrete(18) | Dense |
| ALE | Seaquest | Box(0, 255, (4, 84, 84)) | Discrete(18) | Dense |
| ALE | Venture | Box(0, 255, (4, 84, 84)) | Discrete(18) | Dense |
| Gymnasium-Robotics | AntMaze_UMaze-v5 | Box(-inf, inf, (27,)) | Box(-1, 1, (8,)) | Sparse |

### B.2   Baselines

We designed the following settings for the baseline experiments, and all the subsequent questions were adjusted based on the baselines. Moreover, all the experiments are performed using the proximal policy optimization (PPO) (Schulman et al., 2017) implementation from RLLTE (Yuan et al., 2023).

Table 4: Details of baseline settings.

| Hyperparameter | Value |
|---|---|
| Observation normalization | RMS |
| Reward normalization | RMS |
| Weight initialization | Orthogonal |
| Update proportion | 1.0 |
| with LSTM | False |

### B.3   Details of Questions

Table 5 illustrates the details of the candidates for all questions.

Table 5: Details of candidates for all questions, where **I** is a batch of intrinsic rewards.

| # | Candidate | Detail |
|---|---|---|
| Q1 | Vanilla | obs. = obs. / 255.0, only for image-based observations, else obs. = obs.. |
| | RMS | obs. = $\text{Clip}\left(\frac{\text{obs.}-\text{running mean}}{\text{running std.}}, -5.0, 5.0\right)$ |
| Q2 | Vanilla | $\mathbf{I} = \mathbf{I}$ |
| | RMS | $\mathbf{I} = \frac{\mathbf{I}}{\text{running std}}$ |
| | Min-Max | $\mathbf{I} = \frac{\mathbf{I}-\min(\mathbf{I})}{\max(\mathbf{I})-\min(\mathbf{I})}$ |
| Q3 | 0.01 | Use 1% of the samples to update the intrinsic reward module. |
| | 0.1 | Use 10% of the samples to update the intrinsic reward module. |
| | 0.5 | Use 50% of the samples to update the intrinsic reward module. |
| | 1.0 | Use 100% of the samples to update the intrinsic reward module. |
| Q4 | Vanilla | Fill the input tensor with values drawn from the uniform distribution. |
| | Orthogonal | Fill the input tensor with a (semi) orthogonal matrix. |
| Q5 | Vanilla | Policy network with only convolutional and linear layers. |
| | LSTM | Policy network that includes an LSTM layer. |
| Q6 | Vanilla | $R = E + I$ |
| | Two-head | Value network uses two separate branches for $E$ and $I$. |
| Q7 | Global+Episodic | E3B+RND, E3B+ICM, E3B+RIDE, RE3+RND, RE3+ICM, RE3+RIDE |
| | Global+Global | RND+ICM, RND+RIDE, ICM+RIDE |

## B.4  Best Configurations

Table 6: The best configurations for each intrinsic reward on *SuperMarioBros*.

| Reward | Obs. Norm. | Reward Norm. | Update Prop. | Weight Init. | Memory Required |
|---|---|---|---|---|---|
| ICM | Min-Max | Vanilla | 1.0 | Default | ✗ |
| RND | RMS | Vanilla | 1.0 | Orthogonal | ✗ |
| Disagreement | RMS | Vanilla | 0.5 | Default | ✗ |
| NGU | RMS | Min-Max | 0.1 | Default | ✗ |
| PseudoCounts | RMS | Min-Max | 0.01 | Default | ✓ |
| RIDE | RMS | Min-Max | 0.01 | Default | ✗ |
| RE3 | Min-Max | Min-Max | N/A | Default | ✗ |
| E3B | Min-Max | Min-Max | 0.1 | Orthogonal | ✓ |

Table 7: The best configurations for each intrinsic reward on *MiniGrid-DoorKey-16×16*.

| Reward | Obs. Norm. | Reward Norm. | Update Prop. | Weight Init. | Memory Required |
|---|---|---|---|---|---|
| ICM | RMS | RMS | 1.0 | Orthogonal | ✗ |
| RND | RMS | Vanilla | 0.5 | Orthogonal | ✗ |
| Disagreement | Vanilla | Min-Max | 0.5 | Default | ✗ |
| NGU | RMS | RMS | 0.01 | Orthogonal | ✗ |
| PseudoCounts | RMS | Min-Max | 1.0 | Orthogonal | ✗ |
| RIDE | RMS | Min-Max | 1.0 | Orthogonal | ✗ |
| RE3 | RMS | Min-Max | N/A | Orthogonal | ✗ |
| E3B | RMS | RMS | 1.0 | Orthogonal | ✗ |

Table 8: PPO hyperparameters for *SuperMarioBros*, *MiniGrid*, and *Procgen* games. These remain fixed for all experiments.

| Hyperparameter | SuperMarioBros | MiniGrid | Procgen |
|---|---|---|---|
| Observation downsampling | (84, 84) | (7, 7) | (64, 64) |
| Observation normalization | / 255. | No | / 255. |
| Reward normalization | No | No | No |
| Weight initialization | Orthogonal | Orthogonal | Orthogonal |
| LSTM | No | No | No |
| Stacked frames | No | No | No |
| Environment steps | 10000000 | 10000000 | 25000000 |
| Episode steps | 128 | 32 | 256 |
| Number of workers | 1 | 1 | 1 |
| Environments per worker | 8 | 256 | 64 |
| Optimizer | Adam | Adam | Adam |
| Learning rate | 2.5e-4 | 2.5e-4 | 5e-4 |
| GAE coefficient | 0.95 | 0.95 | 0.95 |
| Action entropy coefficient | 0.01 | 0.01 | 0.01 |
| Value loss coefficient | 0.5 | 0.5 | 0.5 |
| Value clip range | 0.1 | 0.1 | 0.2 |
| Max gradient norm | 0.5 | 0.5 | 0.5 |
| Epochs per rollout | 4 | 4 | 3 |
| Batch size | 256 | 1024 | 2048 |
| Discount factor | 0.99 | 0.99 | 0.999 |

# C   Usage Examples

## C.1   API Compatiblity

The following table provides a detailed algorithm and environment compatibility of the implemented intrinsic rewards. Since NGU, PseudoCounts, RIDE, and E3B require an episode memory for the reward computation, when combined with off-policy RL algorithms, they can only work in a non-vectorized environment currently. Therefore, these reward modules are marked with a ∗ symbol.

Table 9: Algorithm and environment compatibility of the RLeXplore framework.

| Algorithm | Image-based Observation | State-based Observation | Continuous Action | Discrete Action | On-policy RL | Off-policy RL |
|---|---|---|---|---|---|---|
| ICM | ✓ | ✓ | ✓ | ✓ | ✓ | ✓ |
| RND | ✓ | ✓ | ✓ | ✓ | ✓ | ✓ |
| Disagreement | ✓ | ✓ | ✓ | ✓ | ✓ | ✓ |
| NGU | ✓ | ✓ | ✓ | ✓ | ✓ | ✓∗ |
| PseudoCounts | ✓ | ✓ | ✓ | ✓ | ✓ | ✓∗ |
| RIDE | ✓ | ✓ | ✓ | ✓ | ✓ | ✓∗ |
| RE3 | ✓ | ✓ | ✓ | ✓ | ✓ | ✓∗ |
| E3B | ✓ | ✓ | ✓ | ✓ | ✓ | ✓∗ |

## C.2   Workflow of RLeXplore

The following code provides an example when using RLeXplore with on-policy algorithms. At each time step, the agent first observes the vectorized environments before taking actions. Then the environments execute the actions and return the step information, which is processed by the `.watch()` function to extract necessary data for the current intrinsic reward. Finally, the intrinsic rewards will be computed, and the module will updated concurrently at the end of the episode.

```python
# load the library
from rllte.xplore.reward import RE3
# create the reward module
irs = RE3(...)
# reset the environment
obs, infos = envs.reset()
# a rollout storage
rs = RolloutStorage(...)
# training loop
for episode in range(...):
    for step in range(...):
        # sample actions
        actions = agent(obs)
        # step the environment
        next_obs, rwds, terms, truncs, infos = envs.step(actions)
        # get data from the transitions
        irs.watch(obs, actions, rwds, next_obs, terms, truncs, infos)
        ...
    # prepare the samples
    samples = dict(observations=rs.obs, actions=rs.actions,
                   rewards=rs.rewards, terminateds=rs.terminateds,
                   truncateds=rs.truncateds, next_observations=rs.next_obs
    )
    # compute the intrinsic rewards
    ## sync (bool): Whether to update the reward module after the
    ## `compute` function, default is `True`.
    intrinsic_rewards = irs.compute(samples, sync=True)
```

In contrast, the workflow is a bit different when using RLeXplore with off-policy algorithms. As shown in the following example, the intrinsic reward will computed at each time step rather than at the end of each episode. Moreover, the intrinsic reward module will be updated using the same samples for policy updates.

```python
# load the library
from rllte.xplore.reward import RE3
# create the reward module
irs = RE3(...)
# reset the environment
obs, infos = envs.reset()
# training loop
while True:
    # sample actions
    actions = agent(obs)
    # step the environment
    next_obs, rwds, terms, truncs, infos = envs.step(actions)
    # get data from the transitions
    irs.watch(obs, actions, rwds, next_obs, terms, truncs, infos)
    # compute the intrinsic rewards at each step
    ## sync (bool): Whether to update the reward module after the
    ## `compute` function, default is `True`
    intrinsic_rewards = irs.compute(
        samples=dict(observations=obs, actions=actions,
                     rewards=rwds, terminateds=terms,
                     truncateds=terms, next_observations=next_obs),
        sync=False)
    ...
    # update the reward module
    batch = replay_storage.sample()
    irs.update(samples=dict(observations=batch.obs,
                            actions=batch.actions,
                            rewards=batch.rewards,
                            terminateds=batch.terminateds,
                            truncateds=batch.truncateds,
                            next_observations=batch.next_obs)
    )
    ...
```

## C.3   Mixed Intrinsic Reward

The following code example shows how to create a mixed intrinsic reward using two independent intrinsic rewards:

```python
from rllte.env import make_atari_env
from rllte.xplore.reward import Fabric, RE3, ICM

# define the mixed intrinsic reward
class TwoMixed(Fabric):
    def __init__(self, m1, m2):
        super().__init__(m1, m2)

    def compute(self, samples, sync):
        rwd1, rwd2 = super().compute(samples, sync)

        return rwd1 + rwd2

if __name__ == "__main__":
    # env setup
    device = "cuda:0"
    envs = make_atari_env(device=device)
    # create two intrinsic reward functions
    irs1 = ICM(envs, device)
    irs2 = RE3(envs, device)
    # create the mixed intrinsic reward function
    irs = TwoMixed(irs1, irs2)
```

### C.4 RLeXplore with Stable-Baselines3

Stable-Baselines3 (SB3) (Raffin et al., 2021) is one of the most successful and popular RL frameworks that provides a set of reliable implementations of RL algorithms in Python. SB3 provides a convenient callback function that can be called at given stages of the training procedure, the following code example demonstrates how to use RLeXplore in SB3 for on-policy RL algorithms:

```python
class RLeXploreWithOnPolicyRL(BaseCallback):
    """
    Combining RLeXplore and on-policy algorithms from SB3.
    """
    def __init__(self, irs, verbose=0):
        super(RLeXploreWithOnPolicyRL, self).__init__(verbose)
        self.irs = irs
        self.buffer = None

    def init_callback(self, model: BaseAlgorithm) -> None:
        super().init_callback(model)
        self.buffer = self.model.rollout_buffer

    def _on_step(self) -> bool:
        """
        This method will be called by the model after each call to `env.step()`.

        :return: (bool) If the callback returns False, training is aborted early.
        """
        observations = self.locals["obs_tensor"]
        device = observations.device
        actions = th.as_tensor(self.locals["actions"], device=device)
        rewards = th.as_tensor(self.locals["rewards"], device=device)
        dones = th.as_tensor(self.locals["dones"], device=device)
        next_observations = th.as_tensor(self.locals["new_obs"], device=device)

        # get data from the transitions
        self.irs.watch(observations, actions, rewards, dones, dones, next_observations)

        return True

    def _on_rollout_end(self) -> None:
        # prepare the data samples
        obs = th.as_tensor(self.buffer.observations)
        # get the new observations
        new_obs = obs.clone()
        new_obs[:-1] = obs[1:]
        new_obs[-1] = th.as_tensor(self.locals["new_obs"])
        actions = th.as_tensor(self.buffer.actions)
        rewards = th.as_tensor(self.buffer.rewards)
        dones = th.as_tensor(self.buffer.episode_starts)
        print(obs.shape, actions.shape, rewards.shape, dones.shape, obs.shape)
        # compute the intrinsic rewards
        intrinsic_rewards = irs.compute(
            samples=dict(observations=obs, actions=actions,
                         rewards=rewards, terminateds=dones,
                         truncateds=dones, next_observations=new_obs),
```

More detailed code examples can be found in the attached supplementary materials.

### C.5 RLeXplore with CleanRL

CleanRL (Huang et al., 2022b) is an open-source project focused on implementing RL algorithms with clean, understandable, and reproducible code. It aims to make RL more accessible by providing implementations that are simpler and more transparent than those typically found in research papers or larger libraries. The following code example demonstrates how to use RLeXplore in CleanRL for on-policy RL algorithms:

```python
# load the library
```

```python
from rllte.xplore.reward import RE3
# create the reward module
irs = RE3(envs=envs, device=device)
...
# get data from the transitions
irs.watch(observations=obs[step], actions=actions[step],
          rewards=rewards[step], terminateds=dones[step],
          truncateds=dones[step], next_observations=next_obs
          )
...
next_obs = obs.clone()
next_obs[:-1] = obs[1:]
next_obs[-1] = next_obs
# compute the intrinsic rewards
intrinsic_rewards = irs.compute(
    samples=dict(observations=obs, actions=actions,
                 rewards=rewards, terminateds=dones,
                 truncateds=dones, next_observations=next_obs),
    sync=True)
# add the intrinsic rewards to the rewards
rewards += intrinsic_rewards
```

More detailed code examples can be found in the attached supplementary materials.

### C.6 Implementing New Intrinsic Reward Modules

In RLeXplore, all intrinsic reward methods inherit from a base reward class that requires two functions to be implemented: `compute()` and `update()`. The `compute()` function processes a batch of on-policy trajectories to calculate intrinsic rewards and is automatically called prior to the PPO update, while the `update()` function uses the same trajectories to update the associated modules. To integrate a new intrinsic reward method, users only need to create a new script that inherits from the base reward class and implements these two functions. Moreover, many pre-defined network modules (e.g., Atari CNN, ResNet CNN) are readily available for import, allowing users to use the currently implemented intrinsic rewards as templates for their own implementations.

# D    Comparative Analysis of Intrinsic Reward Implementations

This section provides a detailed comparative analysis of our intrinsic reward implementations in the RLeXplore framework against other publicly available implementations. The results are compiled in tables for different environments to demonstrate the performance of each algorithm. We cite the works from which we obtained the original results in each of the tables, and we provide our results by averaging the performance of the last 100 training episodes over 3 seeds.

## D.1    SuperMarioBros (Only Intrinsic Rewards)

Table 10: Comparison of % of level completed in SuperMarioBros without task rewards.

| Algorithm | % of Level Completed (10M Steps) | % of Level Completed (1M Steps) |
|---|---|---|
| (Original) RIDE | - | 23% |
| (Original) ICM | 30% | - |
| (RLeXplore) RIDE | **100%** | **50%** |
| (RLeXplore) ICM | 30% | 2% |

The percentage of the level completed is computed by dividing the episode return by 3,000, which corresponds to the maximum reward that can be obtained in *SuperMarioBros-1-1* (if the agent solves the level without wasting time). Note that in Figure 3, we divide this quantity by 100 and show a maximum reward of 30.

Note that our implementation of ICM reproduces the results reported in the original paper in Mario (Pathak et al., 2017), and our implementation of RIDE further outperforms the original implementation.

## D.2    MiniGrid-DoorKey-16×16 (Extrinsic + Intrinsic Rewards)

Table 11: Episode returns in MiniGrid-DoorKey-16×16 with extrinsic and intrinsic rewards.

| Algorithm | Episode Return (10M Steps) |
|---|---|
| (Original) RIDE (Zhang et al., 2020) | 0.25 |
| (Original) ICM (Zhang et al., 2020) | 0.0 |
| (Original) RND (Zhang et al., 2020) | 0.0 |
| (Original) IMPALA (Zhang et al., 2020) | 0.0 |
| (RLeXplore) PPO | 0.37 |
| (RLeXplore) ICM | **0.6** |
| (RLeXplore) RND | **0.6** |
| (RLeXplore) RIDE | 0.12 |

Using the implementations in RLeXplore we obtain significantly better performance in the same tasks and with the same algorithms.

### D.3  MiniGrid-DoorKey-8×8 (Extrinsic + Intrinsic Rewards)

We also evaluate our implementations in *MiniGrid-DoorKey-8×8* with a budget of 1M environment steps to be able to compare to the original results reported in (Seo et al., 2021).

Table 12: Episode returns in MiniGrid-DoorKey-8×8 with 1M environment steps.

| Algorithm | Episode Return (1M Steps) |
|---|:---:|
| (Original) RE3 (Seo et al., 2021) | 0.5 |
| (Original) RND (Seo et al., 2021) | 0.0 |
| (Original) ICM (Seo et al., 2021) | 0.2 |
| (Original) A2C (Seo et al., 2021) | 0.0 |
| (RLeXplore) RE3 | **0.95** |
| (RLeXplore) RND | **0.82** |
| (RLeXplore) ICM | 0.83 |
| (RLeXplore) PPO | 0.22 |

Importantly, we reproduce the results reported in (Seo et al., 2021) very accurately, showing that RE3 can provide more sample-efficient exploration in this domain, compared to RND and ICM. Still, our implementations of RE3 and ICM achieve even better performance than the original ones.

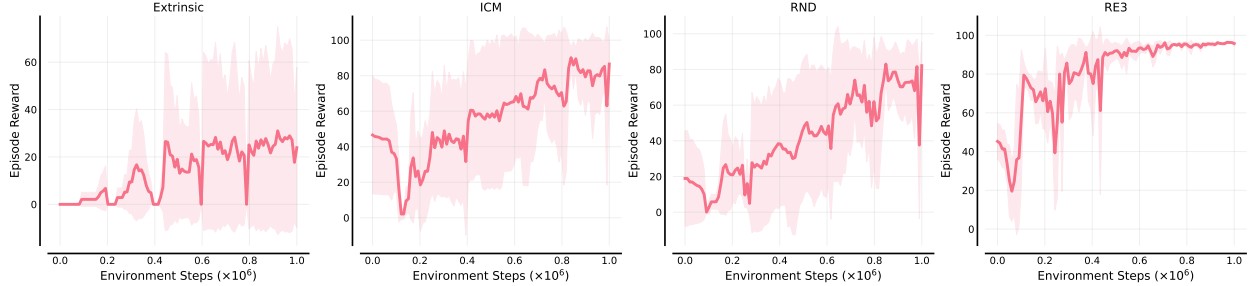

Figure 9: Using RLeXplore in *MiniGrid-DoorKey-8×8*, we are able to not only reproduce the conclusions obtained in previous work (Seo et al., 2021) regarding the capabilities of RE3 compared to ICM and RND, but we also generally achieve better performance, hence providing stronger baselines to the RL community.

### D.4  Procgen - 200 Mazes (Extrinsic + Intrinsic Rewards)

Table 13: Performance comparison in Procgen - 200 Mazes with 25M training steps.

| Algorithm | Procgen - 200 Mazes (25M Steps) |
|---|:---:|
| (Original) E3B (Castanyer et al., 2023) | 3.0 |
| (Original) ICM (Castanyer et al., 2023) | 2.5 |
| (Original) RND (Castanyer et al., 2023) | 1.7 |
| (RLeXplore) E3B | **4.1** |
| (RLeXplore) ICM | **5.9** |
| (RLeXplore) RND | **5.0** |

### D.5  ALE-5 (Extrinsic + Intrinsic Rewards)

In this section, we present the evaluation results of the intrinsic reward methods on a set of ALE games known for their challenging exploration requirements. These "hard-exploration" games, including Gravitar, Montezuma's Revenge, Private Eye, Seaquest, and Venture, serve as a benchmark for testing the effectiveness of intrinsic rewards in aiding exploration and improving agent performance.

We observe that while intrinsic rewards lead to a decline in performance in Gravitar, they generally provide substantial benefits, particularly in environments where exploration is difficult. For example, in Seaquest,

Table 14: Mean performance across different environments for each algorithm, averaged over 3 seeds after 25M environment steps. Results are averaged over the last 100 episodes of training. In Gravitar, intrinsic rewards appear to hinder the performance of the extrinsic agent, whereas, in other environments, they significantly enhance performance. Notably, in Seaquest, the extrinsic agent ranks among the lowest, highlighting the benefit of intrinsic rewards. All experiments were conducted using sticky actions with a repeat probability of 0.25.

| Algorithm | Gravitar | MontezumaRevenge | PrivateEye | Seaquest | Venture |
|---|---|---|---|---|---|
| Extrinsic | **1060.19** | 42.83 | 88.37 | 942.37 | 391.73 |
| Disagreement | 689.12 | 0.00 | 33.23 | 6577.03 | 468.43 |
| E3B | 503.43 | 0.50 | 66.23 | **8690.65** | 0.80 |
| ICM | 194.71 | 31.14 | -27.50 | 2626.13 | 0.54 |
| PseudoCounts | 295.49 | 0.00 | **1076.74** | 668.96 | 1.03 |
| RE3 | 130.00 | 2.68 | 312.72 | 864.60 | 0.06 |
| RIDE | 452.53 | 0.00 | -1.40 | 1024.39 | 404.81 |
| RND | 835.57 | **160.22** | 45.85 | 5989.06 | **544.73** |

the use of intrinsic rewards enables algorithms to significantly outperform the extrinsic agent, which ranks among the lowest.

Note that we do not compare these results to other works because evaluation settings differ significantly between papers. For instance, in our case, we used sticky actions with a probability of 0.25%, which makes the exploration problem more difficult, and it is not always used. Also, we trained our agents for 25M steps instead of the standard 200M due to computational constraints. Still, our results provide evidence that intrinsic rewards are generally helpful in achieving better episode returns in hard-exploration environments.

Table 15: Aggregated performance comparison of mean episode return between RLeXplore and original implementations on the ALE-5 benchmark. The results show average returns across five Atari games. Since the original implementations of R2D2 and NGU are not publicly available, we report their published results obtained with 35B environment steps, while our experiments were conducted with 100 million frames.

| Algorithm | RL Algorithm | Frames | Mean Episode Return |
|---|---|---|---|
| Extrinsic - PPO | PPO | 100M | 0.5k |
| Extrinsic - R2D2 (Kapturowski et al., 2018) | R2D2 | 35B | 210k |
| (RLeXplore) Disagreement | PPO | 100M | 1.5k |
| (RLeXplore) NGU | PPO | 100M | 0.8k |
| (RLeXplore) PseudoCounts | PPO | 100M | 0.4k |
| (Original) Disagreement (Pathak et al., 2019) | PPO | 100M | 0.1k |
| (Original) NGU (Badia et al., 2020) | R2D2 | 35B | 225k |

Table 15 further provides a direct performance comparison between our RLeXplore implementations and the original ones reported in the literature. Since no public codebase or dataset is available for NGU, we extracted its baseline performance numbers directly from the paper. Despite operating under a more limited training budget, our NGU implementation still achieves competitive performance compared to the published results. For Disagreement, we used its official repository and trained the model for 1M frames. The scores we obtained match the results reported in (Pathak et al., 2019), confirming that our implementation faithfully reproduces the expected performance. Overall, the improved performance of RLeXplore's implementations is largely attributable to the extra effort put into fine-tuning low-level details, which helps to optimize the interplay of the various components.

## D.6 Comparison with Other Projects

Table 16 illustrates the details of official implementations of the included intrinsic rewards in RLeXplore. It is natural to find that they are implemented (1) in different codebases with (2) different libraries (e.g., PyTorch vs Tensorflow), (3) using different RL algorithms (PPO, IMPALA, A3C, A2C), and (4) supporting different environments (ALE, Mario, MiniGrid, DMC). These details further motivate the development of a

Table 16: Details on official implementations of the included intrinsic rewards. **Decoupled**: Did the code decouple the intrinsic reward modules from the RL optimization algorithms, which can be directly reused in other projects?

| Reward | Official Repository | ML framework | Backbone RL algorithm | Supported Tasks | Decoupled |
|---|---|---|---|---|---|
| ICM | Repository | Tensorflow | A3C | SuperMarioBros, VizDoom | ✗ |
| RND | Repository | Tensorflow | PPO | ALE | ✗ |
| Disagreement | Repository | Tensorflow | PPO | SuperMarioBros, ALE, Maze | ✗ |
| NGU | N/A | N/A | N/A | N/A | N/A |
| PseudoCounts | from NGU | N/A | N/A | N/A | N/A |
| RIDE | Repository | PyTorch | IMPALA | MiniGrid | ✗ |
| RE3 | Repository | PyTorch | A2C, Dreamer, RAD | DMControl, MiniGrid | ✗ |
| E3B | Repository | PyTorch | IMPALA | MiniHack, VizDoom | ✗ |

unified framework for training RL agents with intrinsic rewards under standardized conditions and reinforce our motivation to develop RLeXplore.

Furthermore, we provide a comparison of the advantages of other popular codebases for training RL agents with intrinsic rewards in terms of the number of intrinsic reward methods implemented, their modularity and ability to reuse components between RL libraries easily, their documentation, and the number of experiments provided. As compared to other existing projects, RLeXplore offers a distinctive advantage by providing a more unified and standardized approach to training RL agents with intrinsic rewards. It allows users to easily swap intrinsic reward modules regardless of RL libraries, which promotes reproducibility and consistency across different research works. Finally, RLeXplore is evaluated on a wide range of benchmarks with over 1,000 experiments, ensuring its reliability and robustness across various scenarios.

Table 17: Comparison between RLeXplore and other reported libraries of intrinsic rewards. Note that we focus on the intrinsic reward methods that are implemented. For instance, CleanRL has many implementations of different RL algorithms, but RND is the only supported intrinsic reward.

| Framework | ML Framework | Number of Algorithms | Plug & Play | Documentation | Benchmark Results |
|---|---|---|---|---|---|
| CleanRL | PyTorch | 1 | ✗ | ✓ | 1 task, 1 experiments |
| DI-Engine | PyTorch | 3 | ✗ | ✓ | 5 tasks, 19 experiments |
| rllib | TensorFlow | 2 | ✗ | ✓ | N/A |
| RLeXplore | PyTorch | 8 | ✓ | ✓ | 17 tasks, over 2000 experiments |

# E Learning Curves

## E.1 Q1

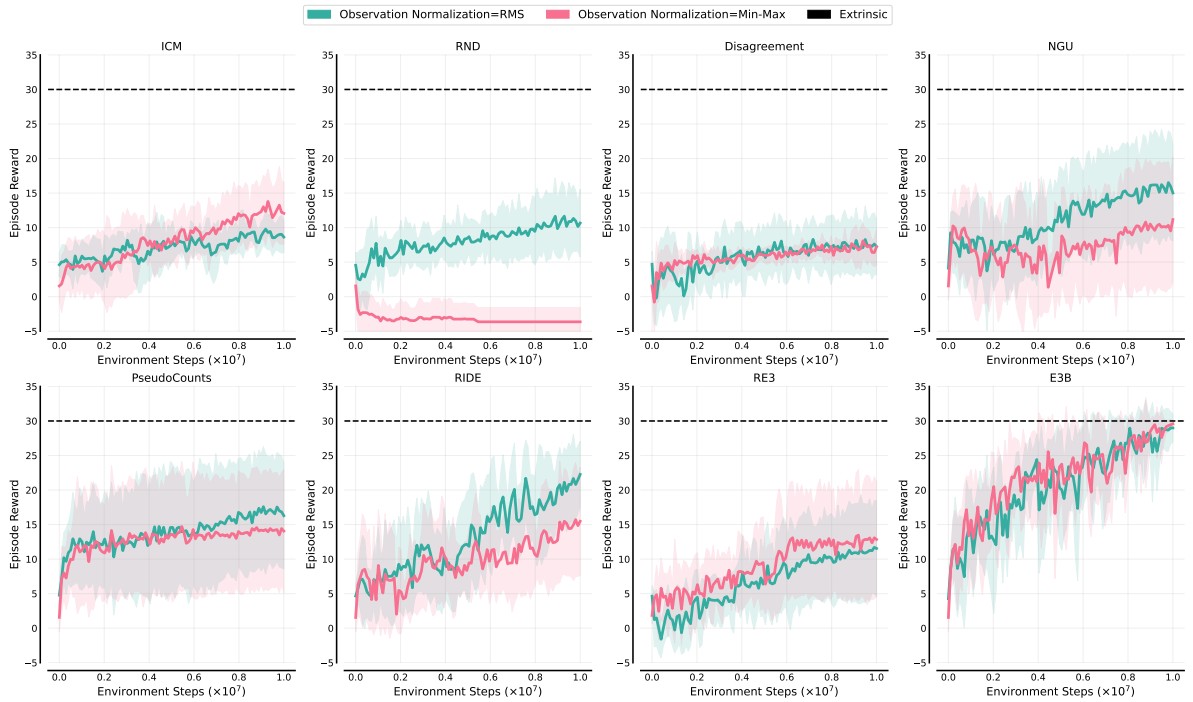

Figure 10: Learning curves of the baselines and Q1 on *SuperMarioBros*. The solid line and shaded regions represent the mean and standard deviation computed with 10 random seeds, respectively.

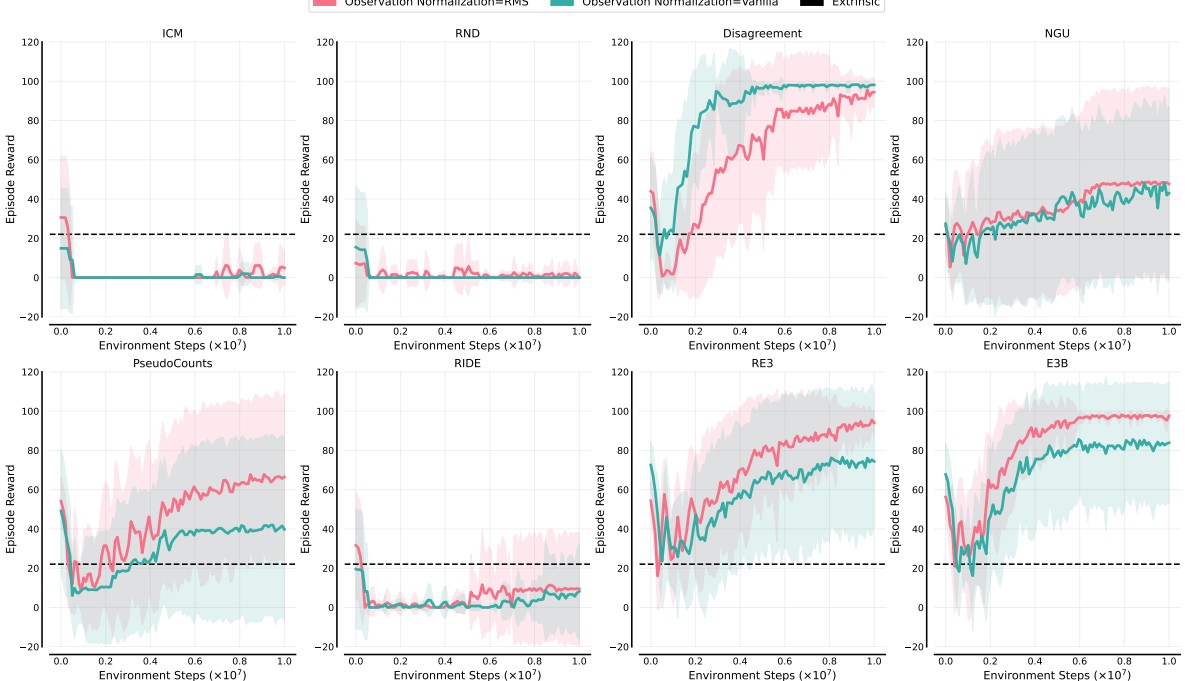

Figure 11: Learning curves of the baselines and Q1 on *MiniGrid-DoorKey-16×16*. The solid line and shaded regions represent the mean and standard deviation computed with 10 random seeds, respectively.

## E.2 Q2

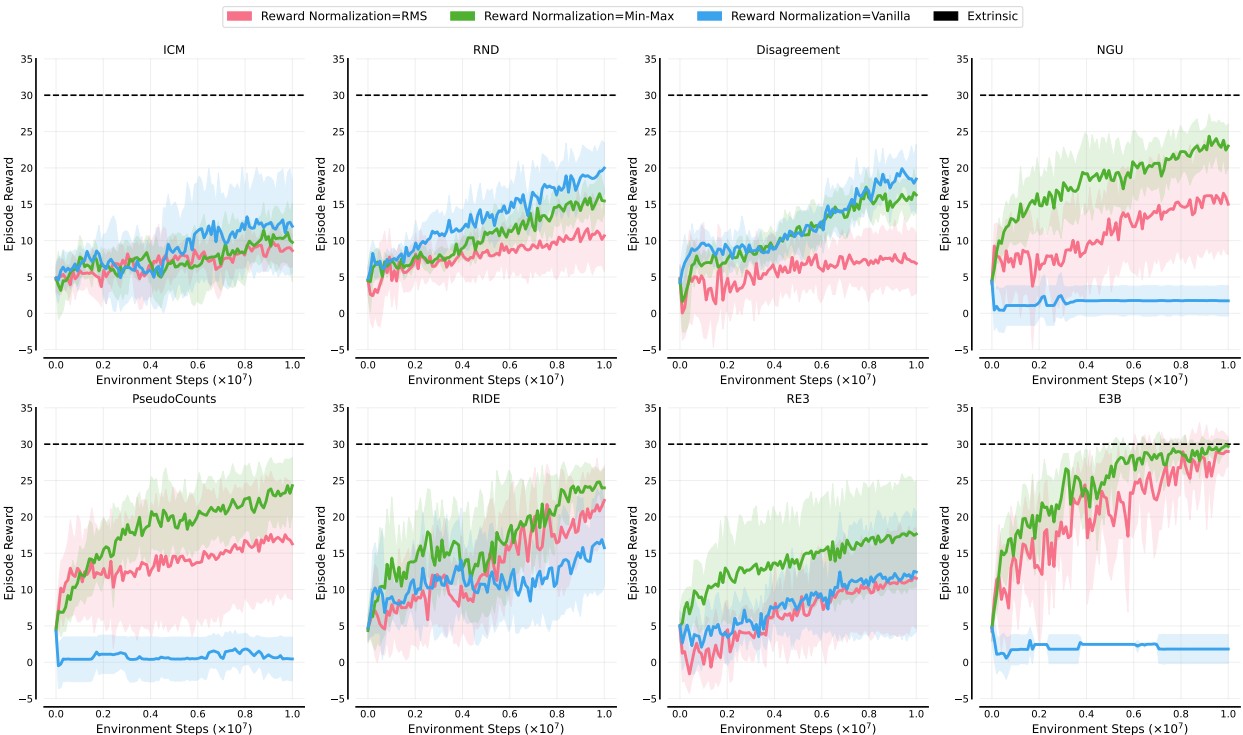

Figure 12: Learning curves of the Q2 on *SuperMarioBros*. The solid line and shaded regions represent the mean and standard deviation computed with 10 random seeds, respectively.

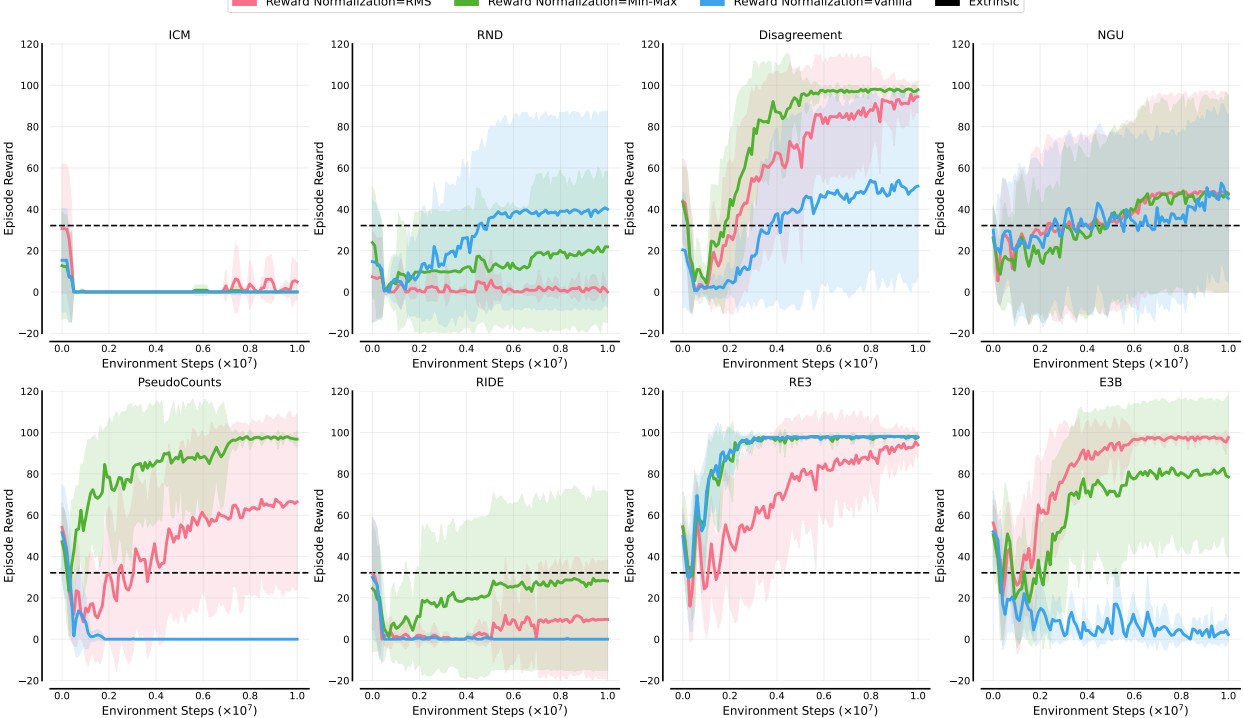

Figure 13: Learning curves of the Q2 on *MiniGrid-DoorKey-16×16*. The solid line and shaded regions represent the mean and standard deviation computed with 10 random seeds, respectively.

## E.3 Q3

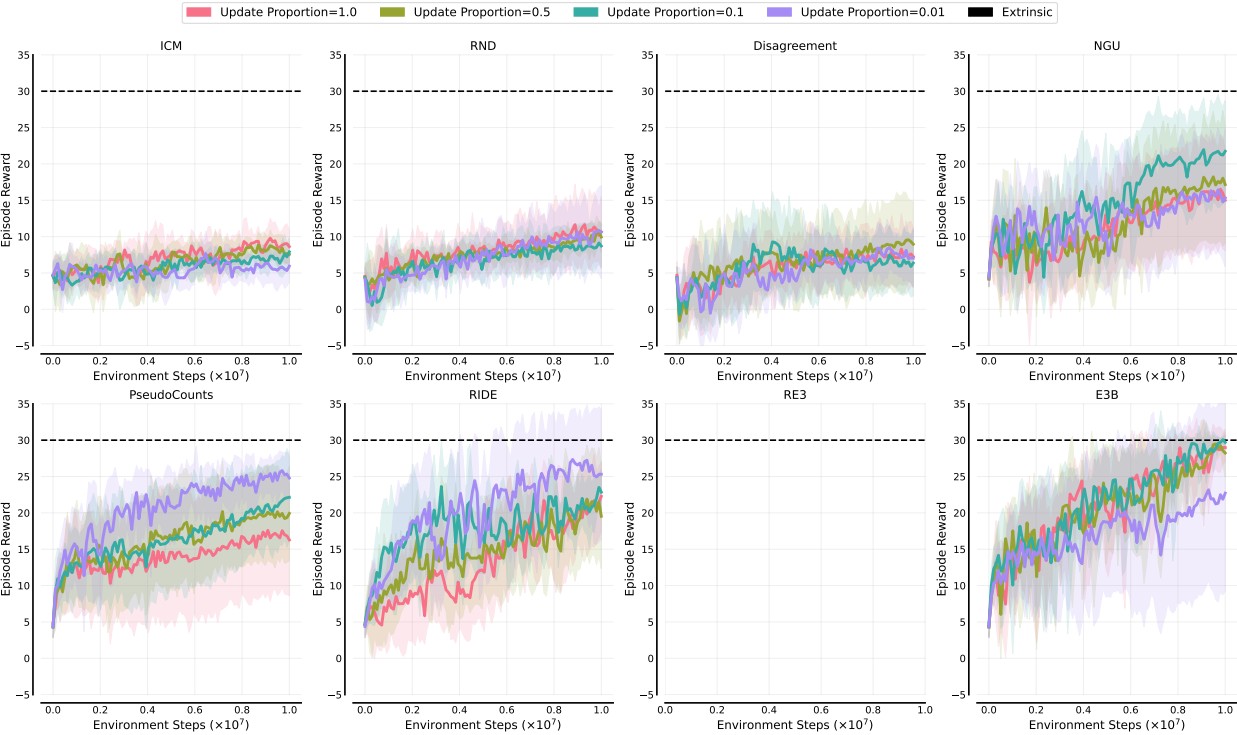

Figure 14: Learning curves of the Q3 on *SuperMarioBros*. The solid line and shaded regions represent the mean and standard deviation computed with 10 random seeds, respectively.

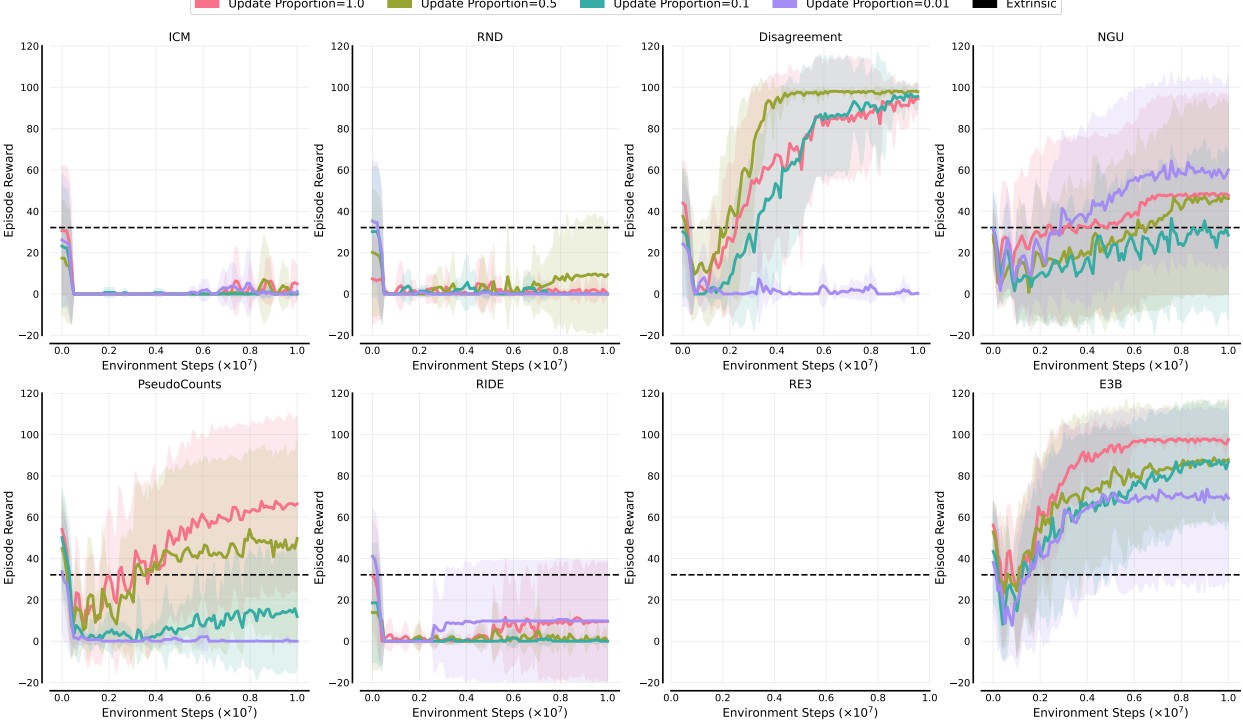

Figure 15: Learning curves of the Q3 on *MiniGrid-DoorKey-16×16*. The solid line and shaded regions represent the mean and standard deviation computed with 10 random seeds, respectively.

## E.4 Q4

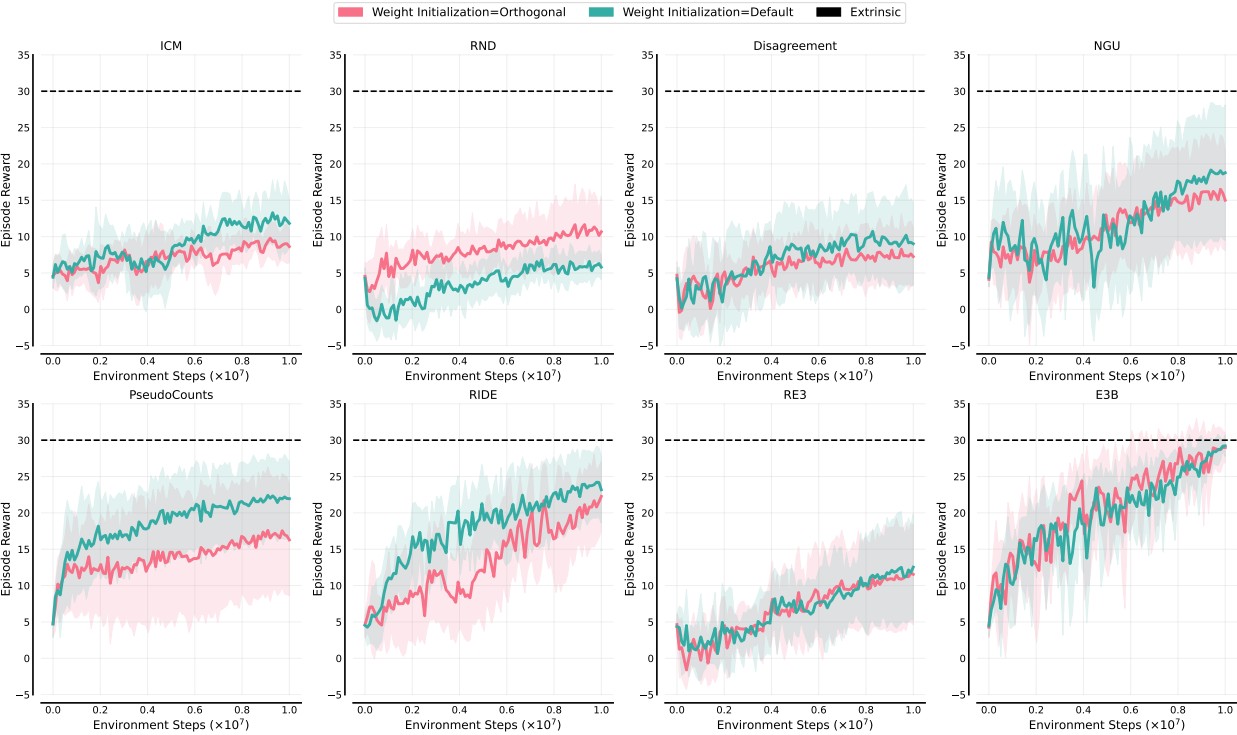

Figure 16: Learning curves of the Q4 on *SuperMarioBros*. The solid line and shaded regions represent the mean and standard deviation computed with 10 random seeds, respectively.

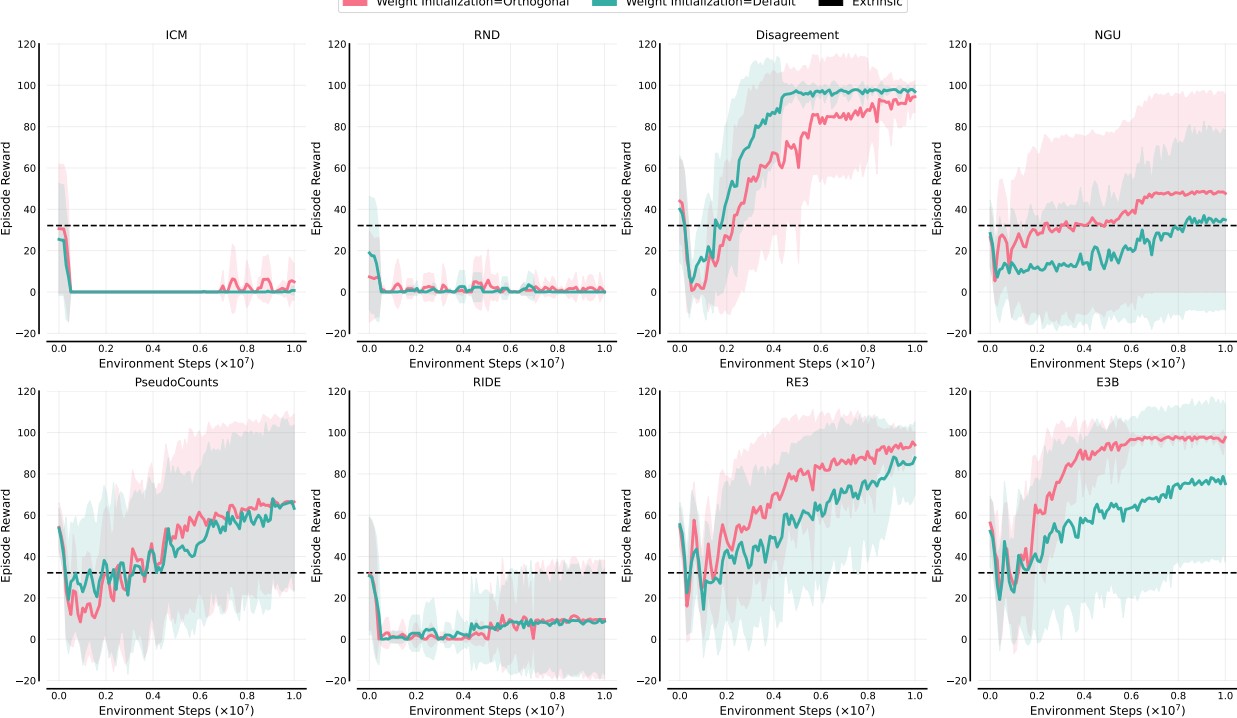

Figure 17: Learning curves of the Q4 on *MiniGrid-DoorKey-16×16*. The solid line and shaded regions represent the mean and standard deviation computed with 10 random seeds, respectively.

## E.5 Q5

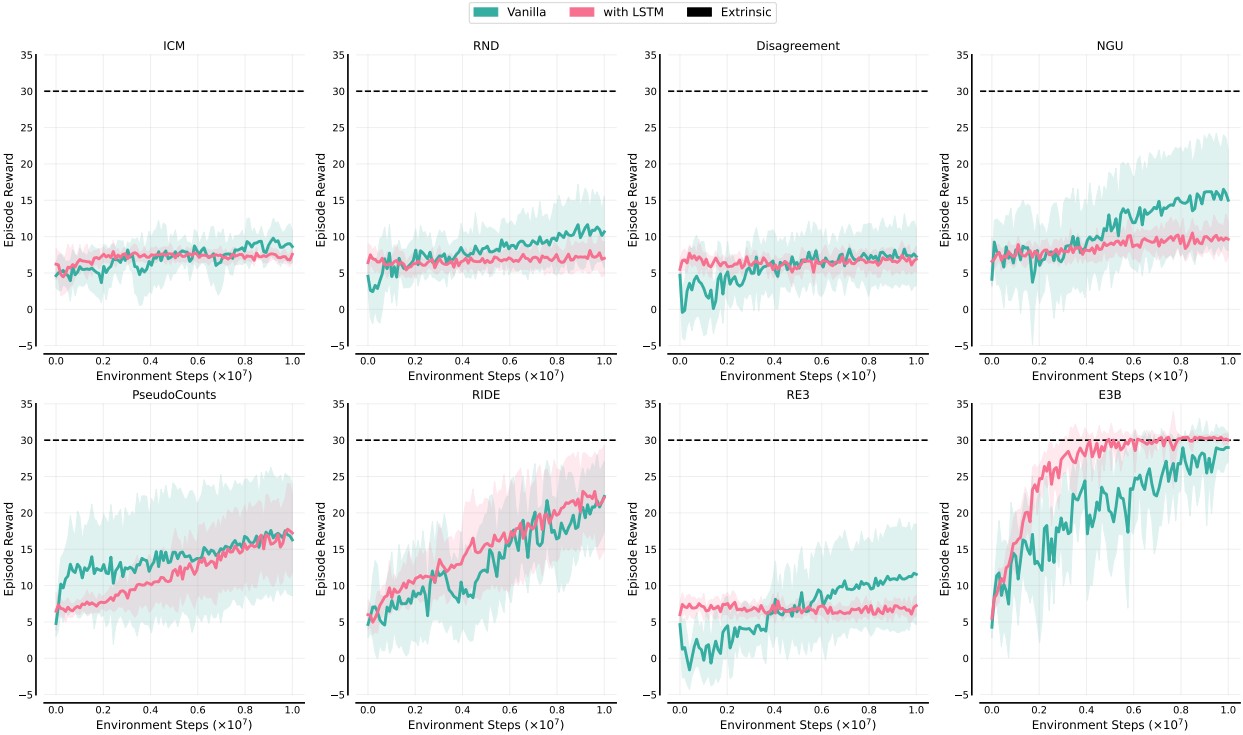

Figure 18: Learning curves of the Q5 on *SuperMarioBros.* The solid line and shaded regions represent the mean and standard deviation computed with 10 random seeds, respectively.

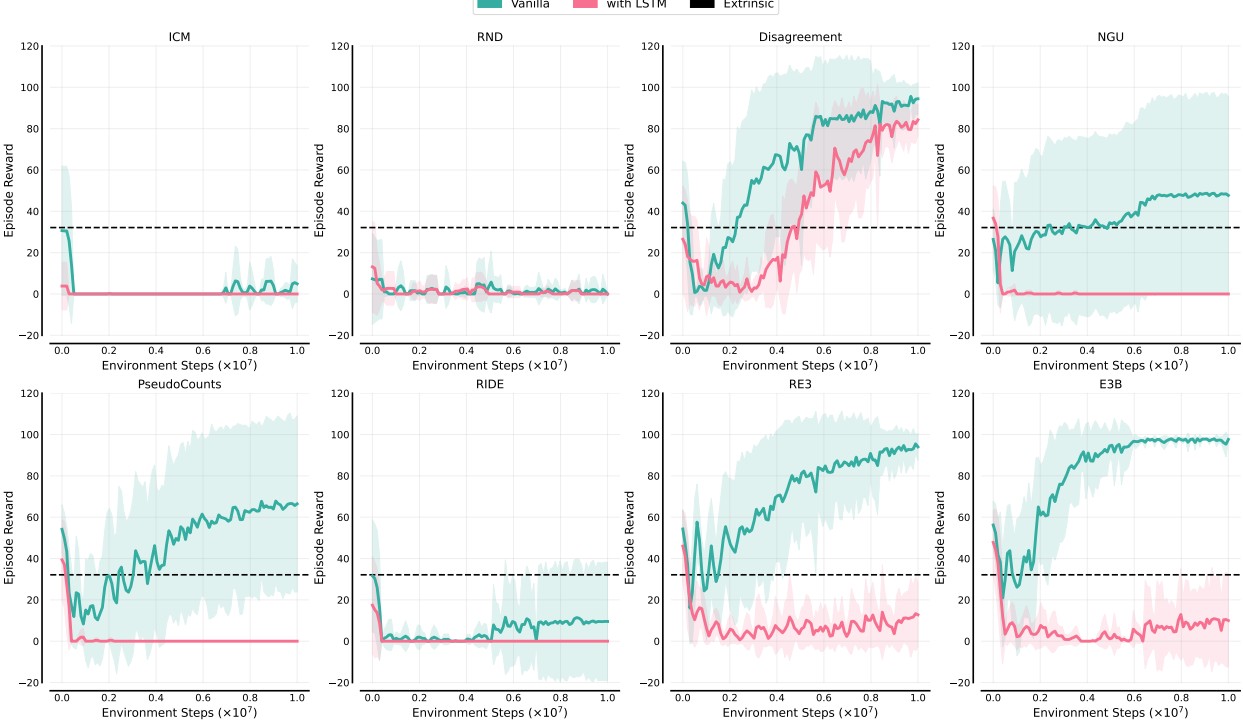

Figure 19: Learning curves of the Q5 on *MiniGrid-DoorKey-16×16.* The solid line and shaded regions represent the mean and standard deviation computed with 10 random seeds, respectively.

## E.6 Q6

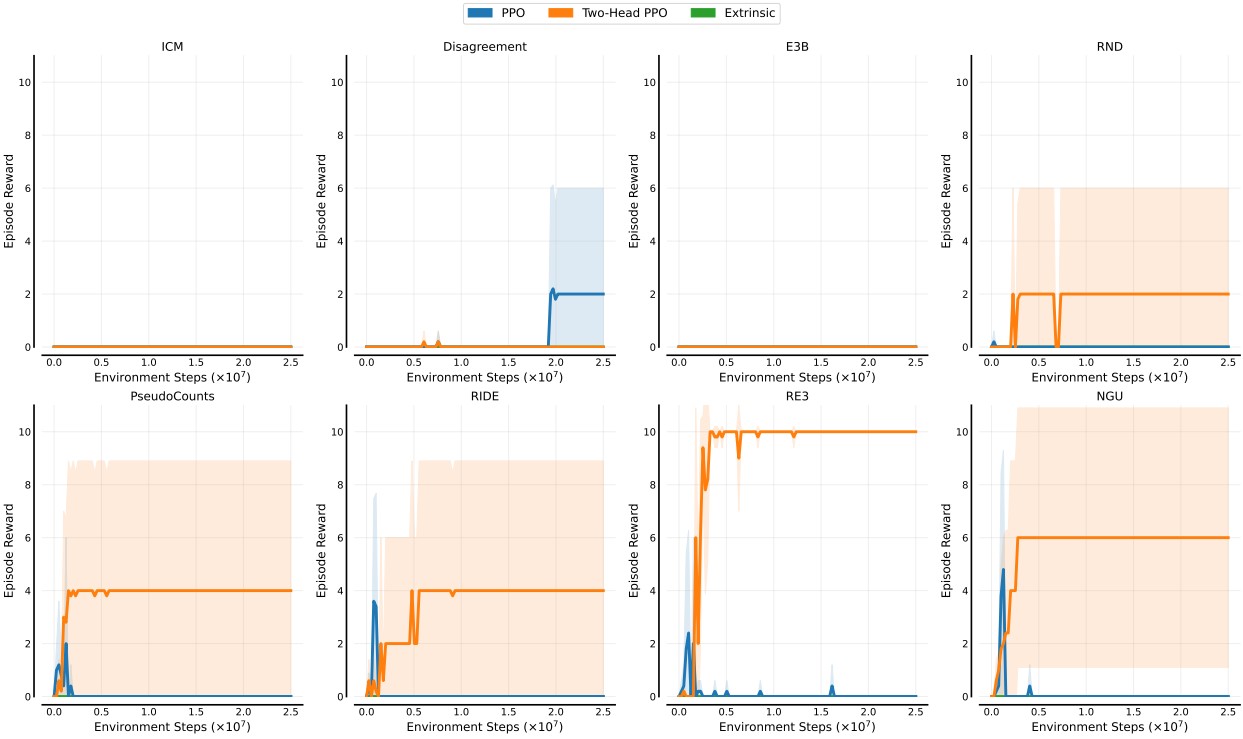

Figure 20: Learning curves of Q6 on *Procgen-1MazeHard*. The solid line and shaded regions represent the mean and standard deviation computed with five random seeds, respectively.

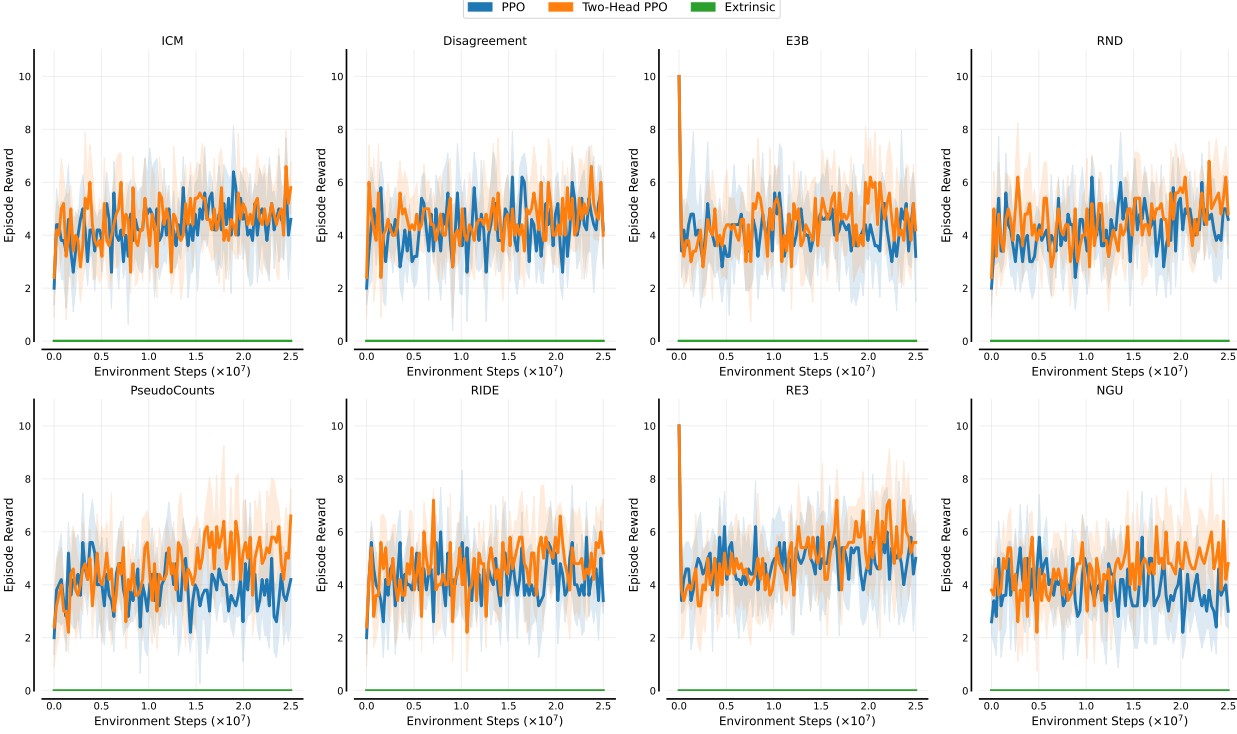

Figure 21: Learning curves of Q6 on *Procgen-AllMazeHard*. The solid line and shaded regions represent the mean and standard deviation computed with five random seeds, respectively.

**E.7    Q7**

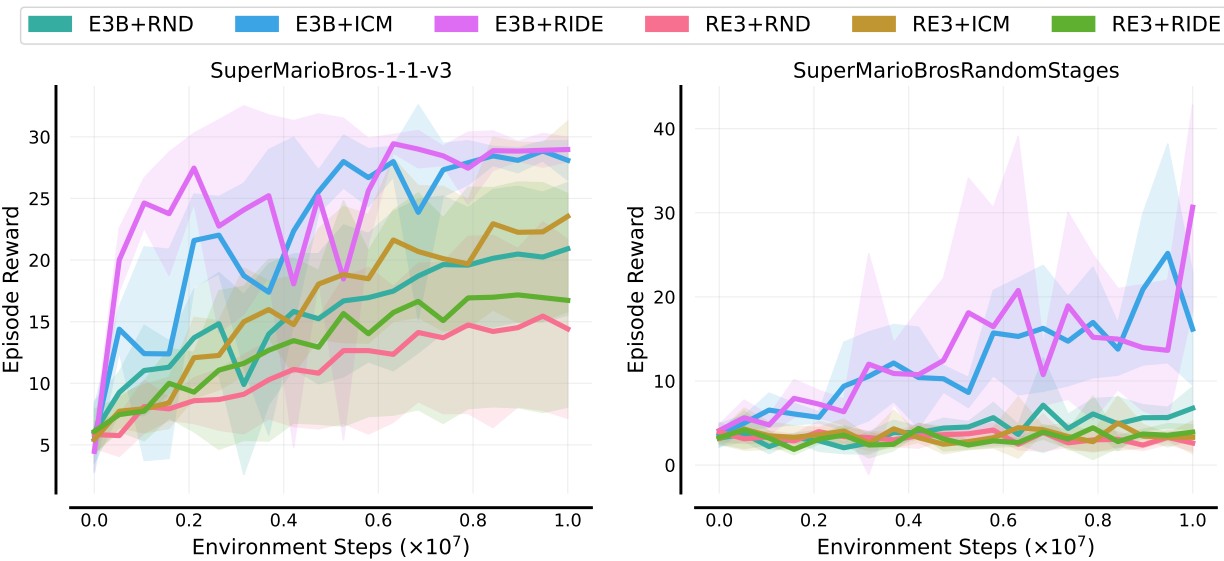

Figure 22: Learning curves of Q7 (global+episodic exploration) on *SuperMarioBros-1-1-v3* and *SuperMarioBrosRandomStages-v3*. The solid line and shaded regions represent the mean and standard deviation computed with five random seeds, respectively.

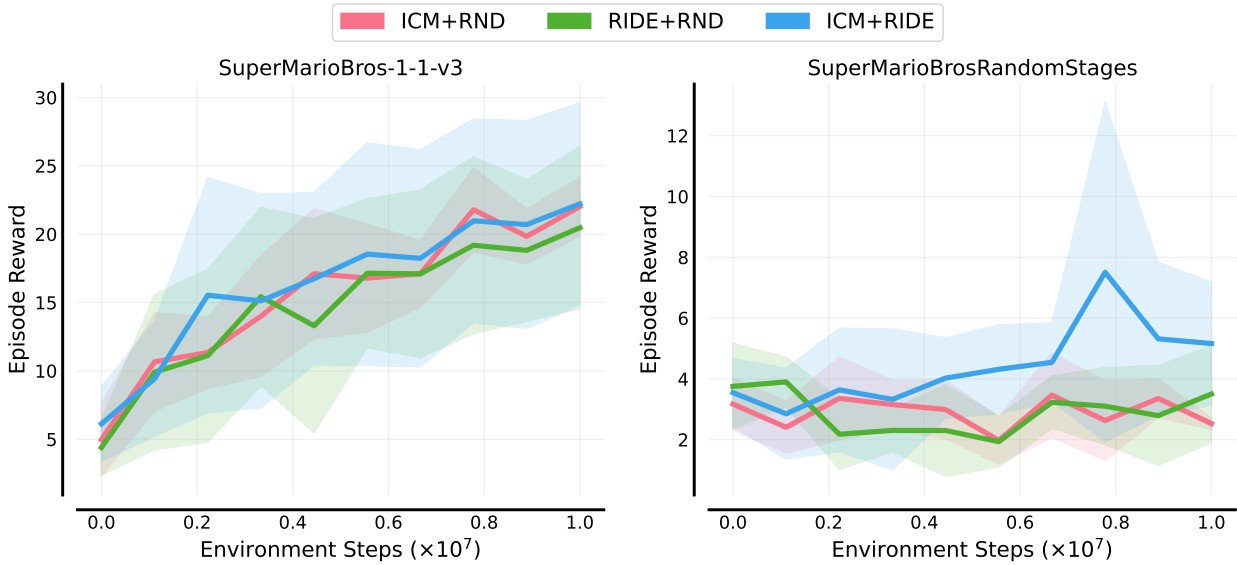

Figure 23: Learning curves of Q7 (global+global exploration) on *SuperMarioBros-1-1-v3* and *SuperMarioBrosRandomStages-v3*. The solid line and shaded regions represent the mean and standard deviation computed with five random seeds, respectively.

## E.8   Additional Experiments for MiniGrid

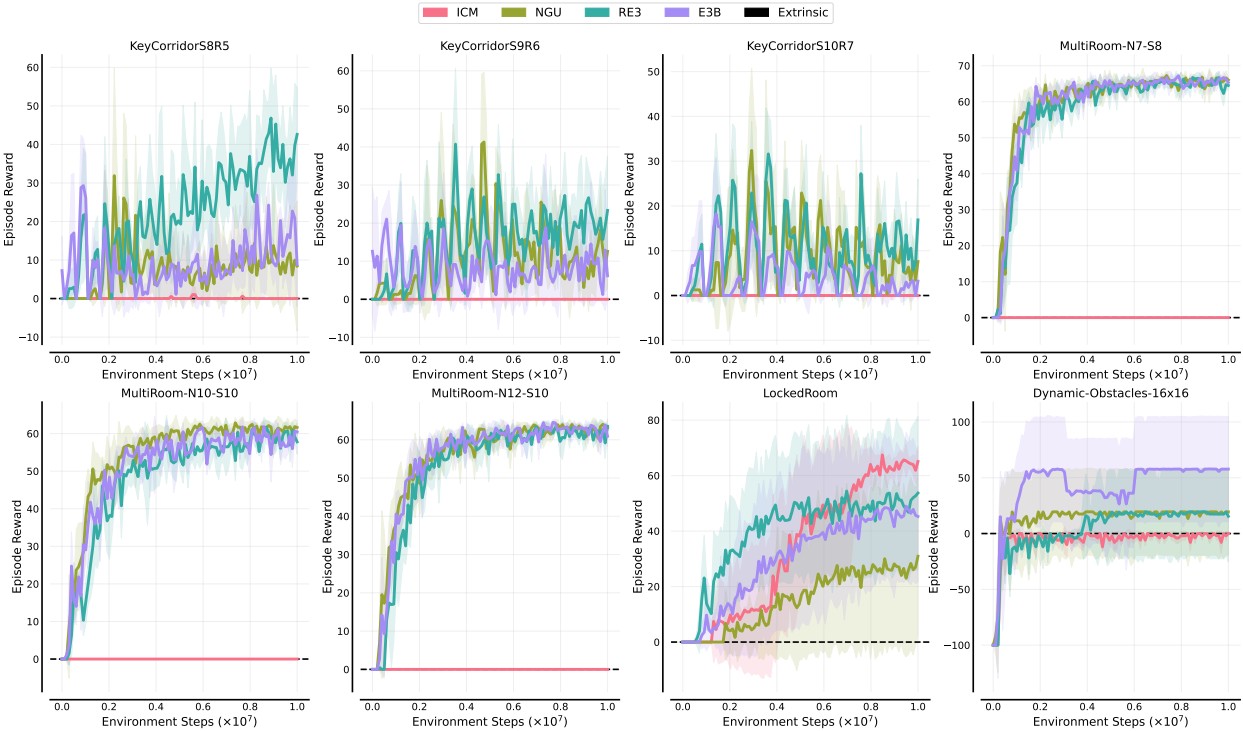

Figure 24: Learning curves of four selected intrinsic rewards on eight extremely hard tasks. The solid line and shaded regions represent the mean and standard deviation computed with five random seeds, respectively.

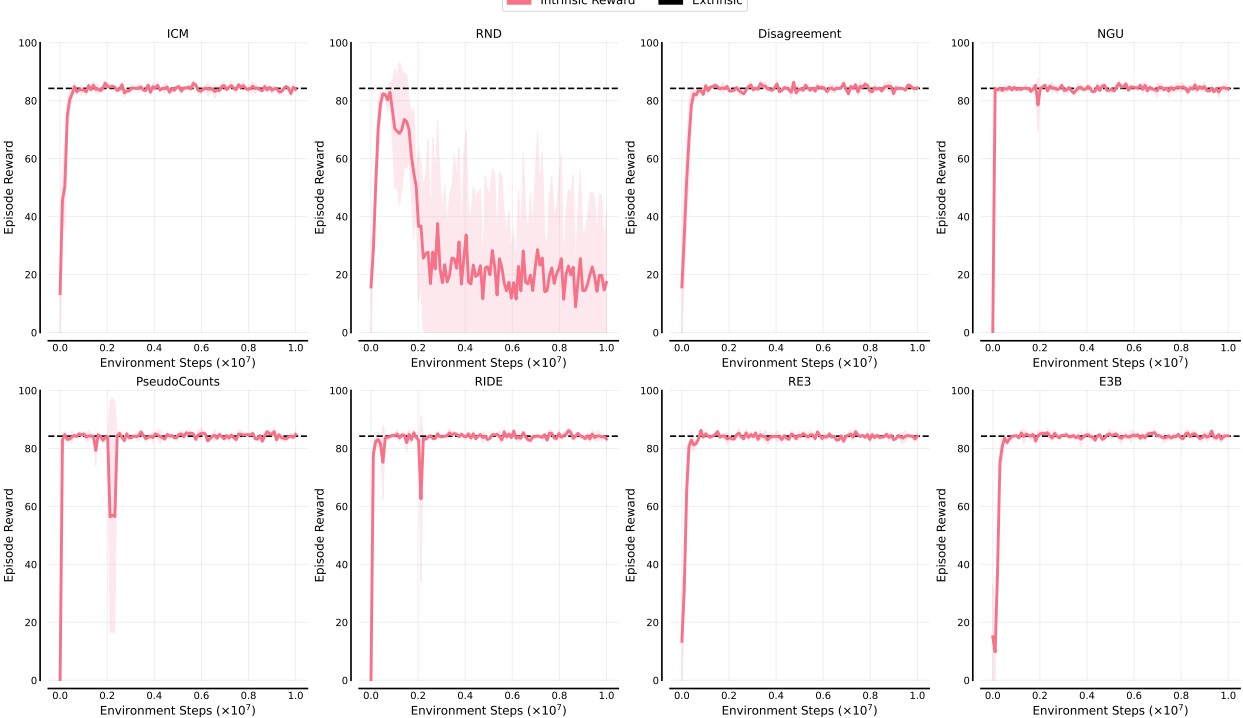

Figure 25: Learning curves on *MiniGrid-MultiRoom-N2-S4-v0*. The solid line and shaded regions represent the mean and standard deviation computed with five random seeds, respectively.

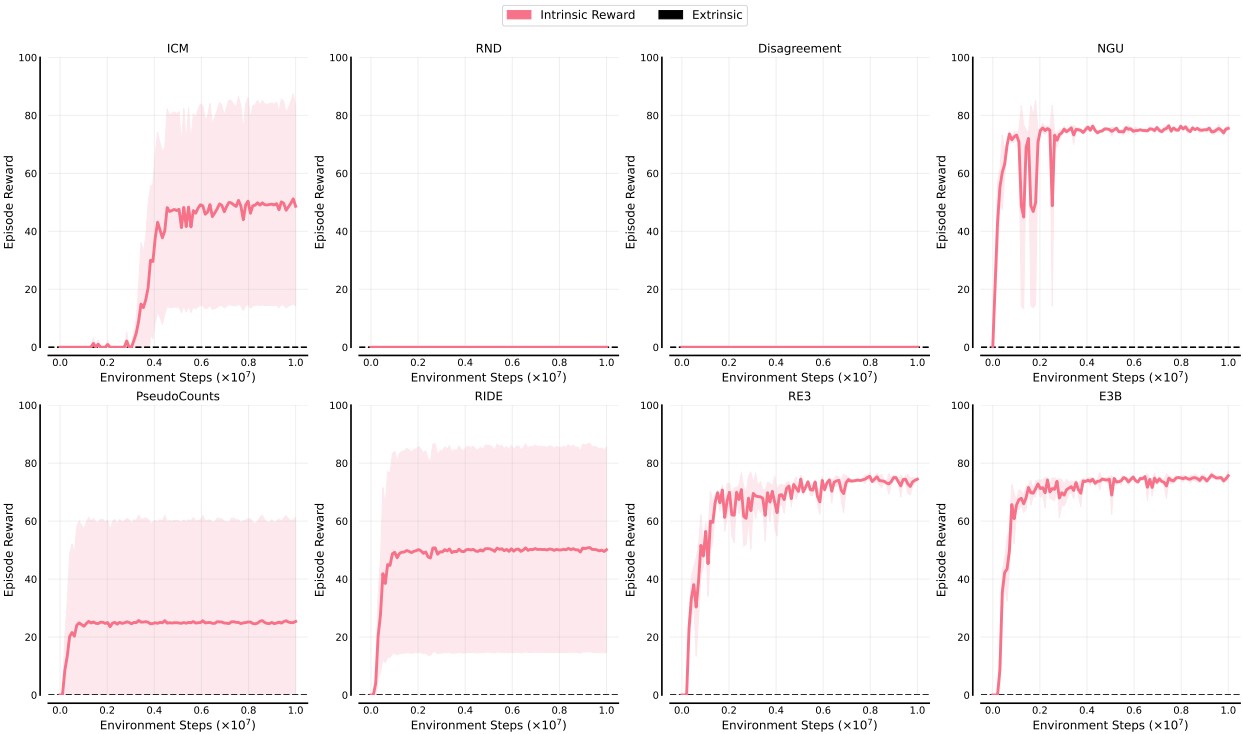

Figure 26: Learning curves on *MiniGrid-MultiRoom-N4-S5-v0*. The solid line and shaded regions represent the mean and standard deviation computed with five random seeds, respectively.

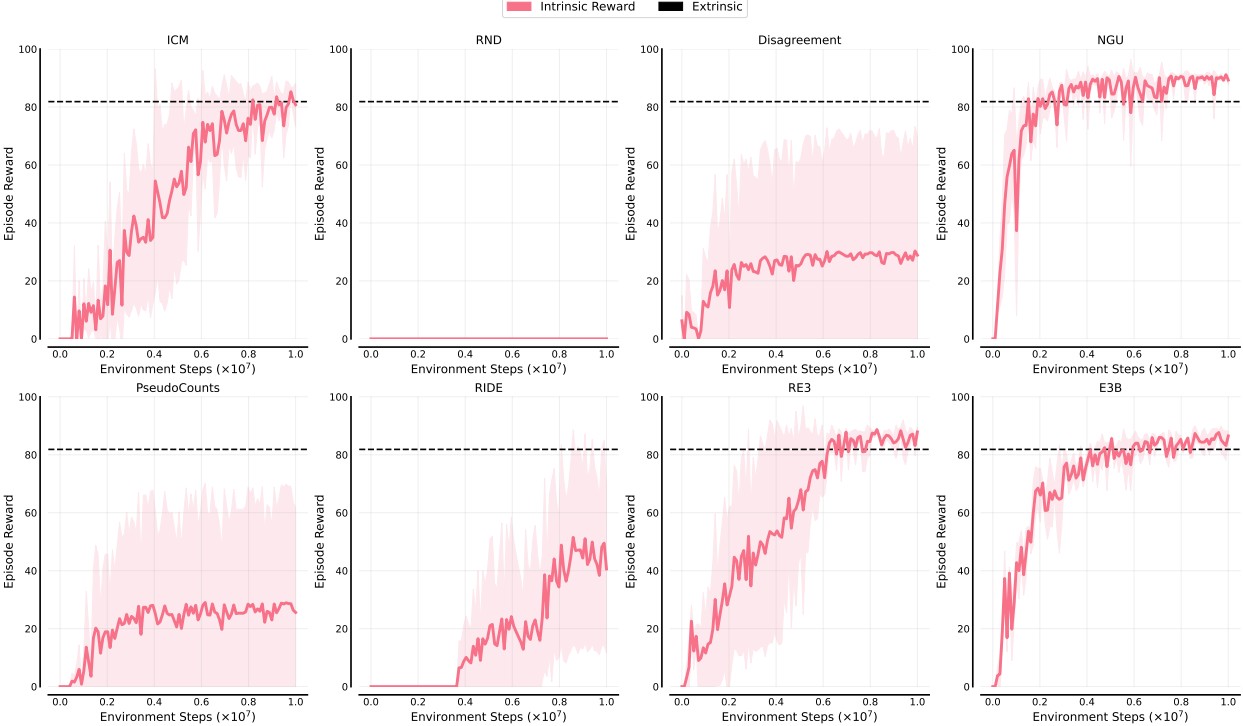

Figure 27: Learning curves on *MiniGrid-KeyCorridorS3R3-v0*. The solid line and shaded regions represent the mean and standard deviation computed with five random seeds, respectively.

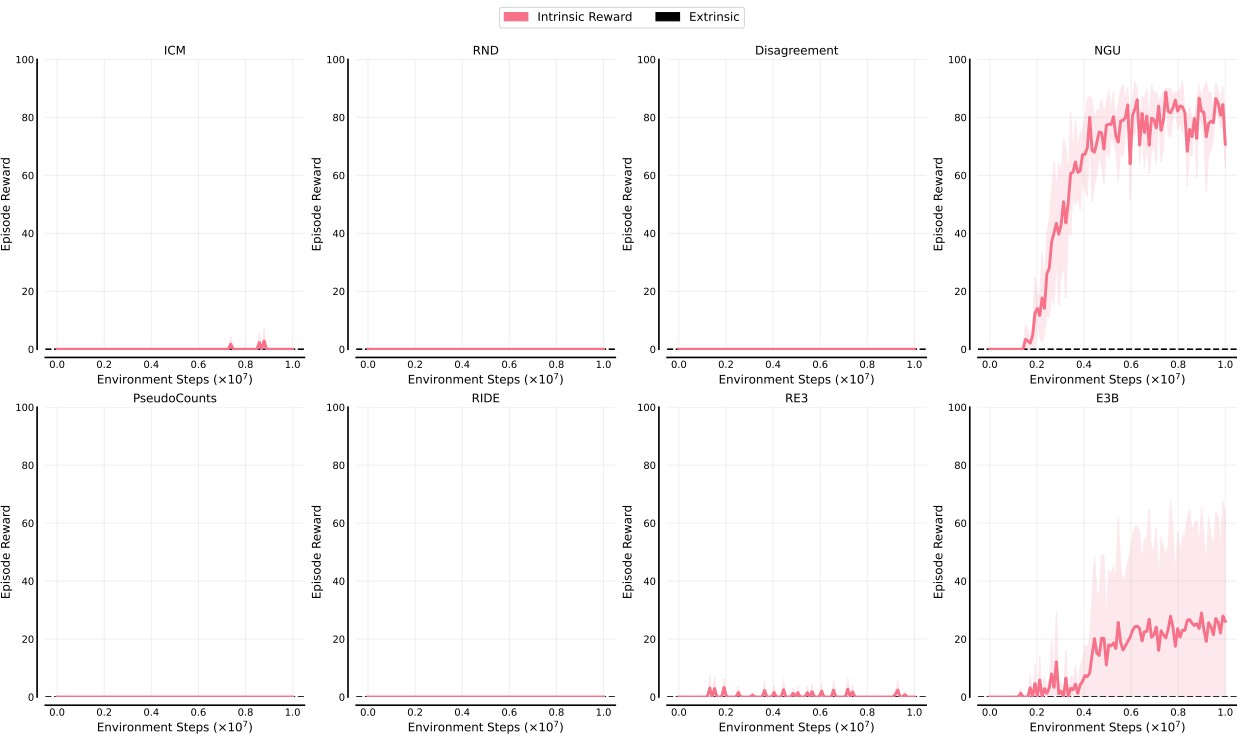

Figure 28: Learning curves on *MiniGrid-KeyCorridorS5R3-v0*. The solid line and shaded regions represent the mean and standard deviation computed with five random seeds, respectively.

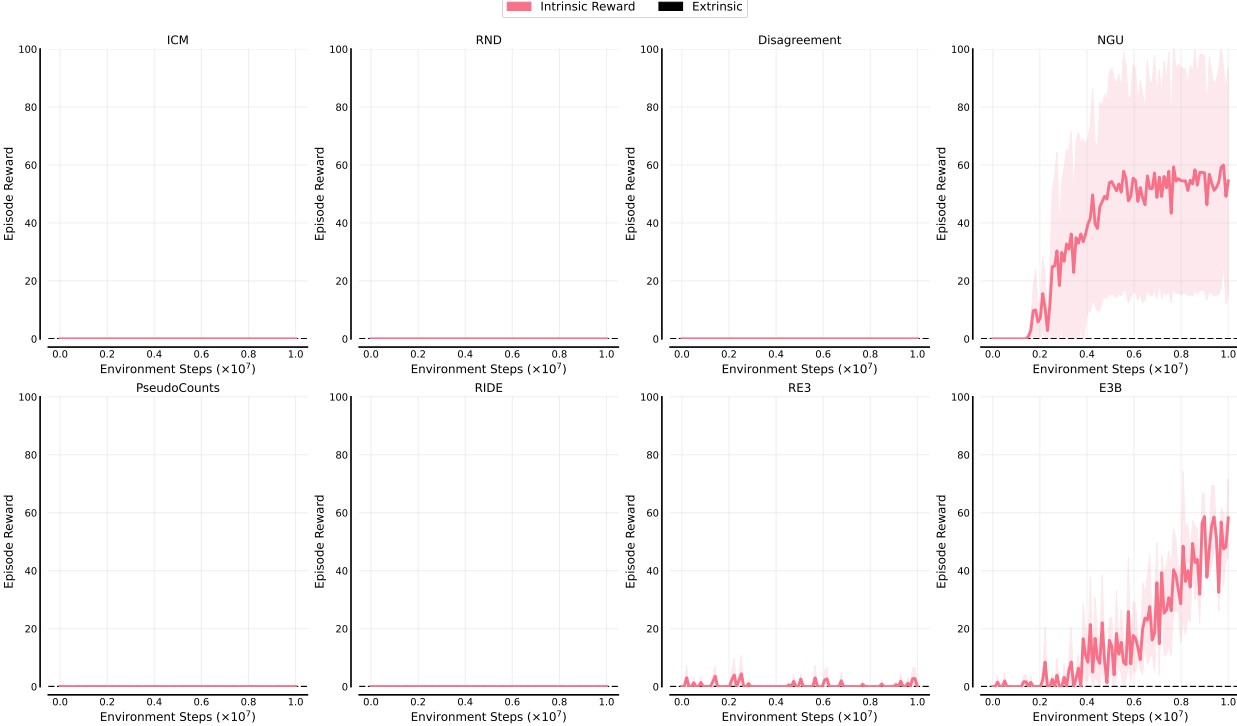

Figure 29: Learning curves on *MiniGrid-KeyCorridorS6R3-v0*. The solid line and shaded regions represent the mean and standard deviation computed with five random seeds, respectively.

# F   On-Policy RL Algorithms and Discrete Control Tasks

In this section, we demonstrate the combination of RLeXplore and on-policy RL algorithms and their effectiveness on discrete control tasks. Specifically, we couple the PPO algorithm and intrinsic rewards and evaluate their performance on *Montezuma Revenge*, a hard exploration task from the ALE benchmark (Bellemare et al., 2013). We use the PPO implementation of CleanRL (Huang et al., 2022b) to show the adaptability of RLeXplore. Table 18 illustrates the training hyperparameters used for the experiments.

Table 18: Training hyperparameters for *Montezuma Revenge*.

| Part | Hyperparameter | Value |
|---|---|---|
| PPO | Observation downsampling | (84, 84) |
| | Stacked frames | 4 |
| | Environment steps | 1e+8 |
| | Episode steps | 128 |
| | Number of workers | 1 |
| | Environments per worker | 8 |
| | Optimizer | Adam |
| | Learning rate | 1e-4 |
| | GAE coefficient | 0.95 |
| | Action entropy coefficient | 0.01 |
| | Value loss coefficient | 0.5 |
| | Value clip range | 0.1 |
| | Max gradient norm | 0.5 |
| | Epochs per rollout | 4 |
| | Batch size | 256 |
| | Discount factor | 0.99 |
| Intrinsic reward | Observation normalization | RMS |
| | Reward normalization | RMS |
| | Weight initialization | Orthogonal |
| | Update proportion | 0.25 |
| | with LSTM | False |

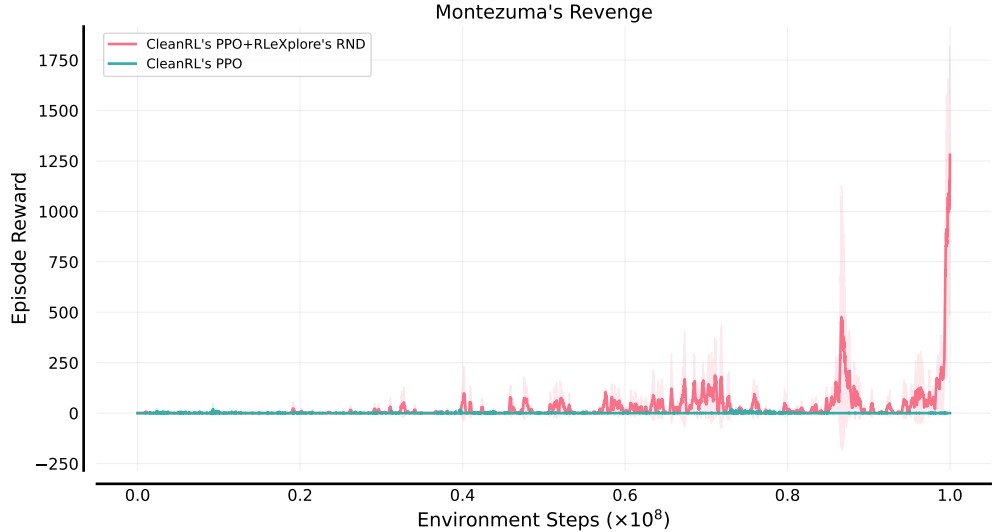

Figure 30: Since only RND can achieve significant results in this task among the eight intrinsic rewards, we only show the results of RND. The solid line and shaded regions represent the mean and standard deviation computed with five random seeds, respectively.

# G   Off-Policy RL Algorithms and Continuous Control Tasks

To showcase the generality of RLeXplore, we run additional experiments in settings different from the ones in the main paper. Concretely, we couple intrinsic rewards with soft actor-critic (SAC) (Haarnoja et al., 2018), an off-policy RL algorithm, and test their performance in *Ant-UMaze*, a continuous control task with sparse rewards. Table 19 illustrates the training hyperparameters used for the experiments. We show the performance of Disagreement, RND, ICM, and vanilla SAC in Figure 31. The results indicate that intrinsically-motivated agents are able to navigate the maze more efficiently, finding the goals more often than the vanilla agents that can only learn from the sparse task rewards.

We only use 3 intrinsic rewards with SAC because of the episodic nature of the other intrinsic reward methods. For example, the episodic memory in RIDE, PseudoCounts, NGU; and the episodic ellipsoid in E3B require the replay buffer to sample entire episodes instead of random rollouts. We aim to implement this logic in our RLeXplore codebase in the future.

Table 19: Training hyperparameters for *Ant-Umaze.*

| Part | Parameter | Value |
|---|---|---|
| SAC | Total timesteps | $1 \cdot 10^6$ |
| | Buffer size | $1 \cdot 10^6$ |
| | Discount ($\gamma$) | 0.99 |
| | Target smoothing coefficient ($\tau$) | 0.005 |
| | Batch size | 256 |
| | Learning starts | 5000 |
| | Policy learning rate | $3 \cdot 10^{-4}$ |
| | Q function learning rate | $1 \cdot 10^{-3}$ |
| | Policy frequency | 2 |
| | Target network frequency | 1 |
| | Noise clip | 0.5 |
| | Entropy coefficient ($\alpha$) | 0.2 |
| | Auto-tune entropy coefficient | True |
| Intrinsic reward | Observation normalization | RMS |
| | Reward normalization | RMS |
| | Weight initialization | Orthogonal |
| | Update proportion | 0.25 |
| | with LSTM | False |

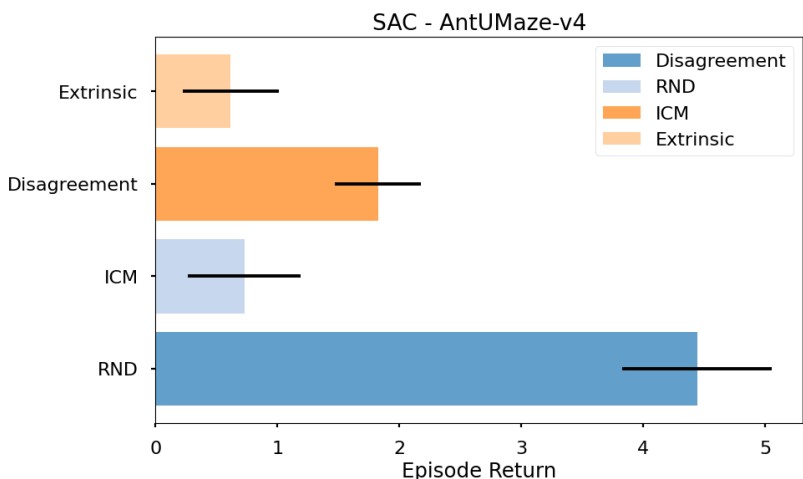

Figure 31: Performance comparison between the three selected intrinsic rewards and the extrinsic reward.

