# OpenReview forum: "RLeXplore: Accelerating Research in Intrinsically-Motivated Reinforcement Learning"
_TMLR — Accepted by TMLR_

### Review · Reviewer_dYVt · 2024-12-05

**Summary Of Contributions:**

The authors introduce RLeXplore, a modular framework that can be used by researchers and practitioners who work on intrinsically motivated reinforcement learning.
- They develop an open-source library that includes eight state-of-the-art intrinsically motivated reinforcement learning techniques.
- They design RLeXplore in a way that can be easily integrated with other RL frameworks.
- Their framework provides a Fabric class that can be used to combine intrinsic rewards from different techniques.
- They provide a systematic experimental analysis of how various implementation details affect the performance of the different intrinsic reward methods.
- In their results, they demonstrate that by carefully choosing the design configurations, they can outperform previous implementations of the same methods.

**Audience:**

Yes

**Claims And Evidence:**

Yes

**Requested Changes:**

I would like to ask the authors to address the aforementioned weakness and to clarify and discuss the open questions/points.

**Strengths And Weaknesses:**

### Strengths:

- The paper is well-written and is easy to follow.
- The need for a standardized framework in intrinsic reward reinforcement learning is clearly presented and is well-motivated.
- The experiments are detailed and comprehensive.
- RLeXplore is designed to encourage the experimentation of new intrinsic reward methods without too much overhead.

### Weaknesses:

- Although the authors have already included enough baselines, the design of RLeXplore does not allow the implementation of unsupervised reinforcement learning methods that first explore and discover skills and then learn them. The authors recognize this limitation in conclusion. However, it can be said that this limits the range of methods that can be implemented with RLeXplore. Therefore, it is worthwhile to discuss and provide some alternative design choices.

Additional points for discussion and clarification:
- What is the purpose of Figure 1 in the introduction? Could you please elaborate on that?
- In Table 1, is the performance increase attributed only to the selection of the different hyperparameters (implementation details). Are the environments exactly the same as the original?
- I find the presentation of the results in Figure 4 confusing and not easy to follow. More specifically, I expected the legend of Figure 4 to be similar to the candidates in Table 5 of the Appendix. For example, in Q2, the approaches are Vanilla, RMS, and Min-Max, whereas in the plot, there is only Min-Max and Vanilla. I understand that some configurations are already covered in Baseline and Combined, but this way of presenting the results is hard to follow.
- Could you clarify why Min-Max reward normalization is better than Combined for SMB in Figure 5? As I understood, Combined is the selection of the best hyperparameters across all questions.
- “Interestingly, in MGD, Min-Max normalization seems to decrease the performance of Disagreement and NGU”. Could you specify the plot to which this statement is referring?
- I would recommend the authors to add some implementation details on the two separate value functions, since this part is vague. For example, are both value functions used in the advantage estimation of PPO?

---

> ### Author Response · Authors · 2025-03-13
> **Response to Reviewer dYVt (Part 1)**
>
> Dear Reviewer,
>
> We sincerely thank you for your insightful and constructive feedback, which has significantly improved the quality and clarity of our manuscript. For easy comparison, in the revision, we use the **\textcolor{magenta}{}** to mark the changes and the **\sout{}** command to indicate the parts that need to be deleted.
>
> **Q1) Although the authors have already included enough baselines, the design of RLeXplore does not allow the implementation of unsupervised reinforcement learning methods that first explore and discover skills and then learn them. The authors recognize this limitation in conclusion. However, it can be said that this limits the range of methods that can be implemented with RLeXplore. Therefore, it is worthwhile to discuss and provide some alternative design choices.**
>
> We intentionally scope RLeXplore to benchmark end-to-end intrinsic reward methods, which optimize intrinsic and extrinsic rewards concurrently. End-to-end methods are more commonly used, so we focus on these algorithms to support a broader portion of the community. Skill-based methods, by contrast, require distinct phases for unsupervised skill discovery and subsequent skill learning (often with a meta-controller), representing a more complex approach.
>
> For readers interested in unsupervised RL benchmarks that incorporate skill-based methods, we refer to Laskin et al. (2021), which offers an alternative framework supporting such approaches.
>
> We have added the following clarification in the revised manuscript:
>
> “RLeXplore is designed to benchmark end-to-end intrinsic reward methods. These end-to-end methods are more commonly used and under-evaluated by the community. Skill-based algorithms, which typically involve separate phases for skill discovery and skill learning, are more complex and left for future work. For an alternative perspective that includes skill-based approaches, we refer readers to the unsupervised RL benchmark by Laskin et al. (2021).“
>
> **Q2) What is the purpose of Figure 1 in the introduction? Could you please elaborate on that?**
>
> Thank you for your feedback. Figure 1 in the introduction was originally used to quickly highlight the contributions of the work, however, we have relocated Figure 1 from the introduction to the experiments section. In its new location, the figure is better contextualized with detailed information about the prior methods, the algorithms being evaluated, the environments, and the claims under investigation.
>
> **Q3) In Table 1, is the performance increase attributed only to the selection of the different hyperparameters (implementation details). Are the environments exactly the same as the original?**
>
> The environments remain exactly the same as in the original work. The performance gains are primarily a result of improvements in our implementation. However, as noted in the paper, a portion of the gains may also stem from using newer RL algorithms to optimize the intrinsic rewards from prior work. For instance, although we mainly use PPO with fixed hyperparameters across most objectives, some objectives were originally optimized using other algorithms like IMPALA or A3C. We provide a detailed discussion of these factors in Appendix D.6.
>
> **Q4) I find the presentation of the results in Figure 4 confusing and not easy to follow. More specifically, I expected the legend of Figure 4 to be similar to the candidates in Table 5 of the Appendix. For example, in Q2, the approaches are Vanilla, RMS, and Min-Max, whereas in the plot, there is only Min-Max and Vanilla. I understand that some configurations are already covered in Baseline and Combined, but this way of presenting the results is hard to follow.**
>
> Thanks for your suggestion. We’ve modified Figure 4 by adding the specific configuration of each RQ in the legend bar, which now aligns with the candidates in Table 5.

---

> ### Author Response · Authors · 2025-03-13
> **Response to Reviewer dYVt (Part 2)**
>
> **Q5) Could you clarify why Min-Max reward normalization is better than Combined for SMB in Figure 5? As I understood, Combined is the selection of the best hyperparameters across all questions.**
>
> The key difference is that Min-Max reward normalization is a specific technique to scale the rewards, while the Combined approach refers to using the best hyperparameter settings identified separately for different aspects of the agent.
>
> In our experiments, we tuned one hyperparameter at a time to clearly see its effect. This allowed us to observe that Min-Max normalization consistently provided stable reward scales. On the other hand, even though the Combined method picks the best individual settings, these hyperparameters can interact in unexpected, non-linear ways. When they are applied together, their interactions may disrupt the delicate balance needed for optimal RL performance, leading to poorer results in the SMB environment.
> Thus, the improved performance with Min-Max normalization likely comes from its consistent and stable reward scaling, whereas the Combined approach suffers from adverse interactions between hyperparameters when merged together.
>
>
> **Q6) "Interestingly, in MGD, Min-Max normalization seems to decrease the performance of Disagreement and NGU”. Could you specify the plot to which this statement is referring?**
>
> We corrected a minor bug in our Min-Max normalization computation (a bug that was identified by a user on GitHub) that was causing this issue. We have since removed this sentence from the manuscript.
>
> **Q7) I would recommend the authors to add some implementation details on the two separate value functions, since this part is vague. For example, are both value functions used in the advantage estimation of PPO?**
>
> We compute separate Generalized Advantage Estimation (GAE) values for the intrinsic and extrinsic rewards. Specifically, we learn two separate value functions - one for intrinsic rewards and one for extrinsic rewards - rather than using a single value function with different n-step return estimators. This allows for more precise credit assignment for each reward type.
>
> The complete paragraph describing this implementation detail is now the following:
>
> "In sparse-reward environments, the objective is for agents to explore the state space by optimizing intrinsic rewards until they discover the task rewards, at which point they should focus solely on optimizing the task rewards. However, many intrinsically motivated RL applications naively optimize the sum of intrinsic and extrinsic rewards, potentially leading to learning fuzzy value functions and suboptimal policies [1]. In this section, we compare this common approach with learning two separate value functions, one for each reward function [2]. The advantages of the latter include the ability to disentangle the effects of intrinsic and extrinsic rewards on the agent's behavior, leading to cleaner learning dynamics and potentially more efficient exploration. In these settings, both value functions are used during the advantage estimation phase of PPO. Specifically, we compute separate GAE values - one using the intrinsic value function and one using the extrinsic value function. The resulting advantages are then summed to compute the policy loss term for PPO. This separation facilitates more accurate advantage estimates for each reward type, leading to improved learning dynamics."
>
> We have now also added a citations [1,2]:
>
> [1] Castanyer R C, Romoff J, Berseth G. Improving Intrinsic Exploration by Creating Stationary Objectives[C]//The Twelfth International Conference on Learning Representations.
>
> [2] Burda Y, Edwards H, Storkey A, et al. Exploration by random network distillation[C]//Seventh International Conference on Learning Representations. 2019: 1-17.
>
> Hope that these can help address the raised comments. If you have any further comments, please don't hesitate to let us know. Thanks!

---

### Review · Reviewer_K5wK · 2025-02-15

**Summary Of Contributions:**

This paper introduces RLeXplore, a unified, modular, and open-source framework designed to facilitate research in intrinsically-motivated reinforcement learning (RL). RLeXplore offers clean implementations of eight state-of-the-art intrinsic reward algorithms, addressing the lack of standardization and insufficient exploration of implementation details in the field. The authors conduct an in-depth study based on 7 research questions to identify critical implementation details and aim to establish standard practices in intrinsically-motivated RL. The framework also encourage the community to enable fair comparisons, easy integration with various RL frameworks, and streamlined development of new intrinsic reward algorithms.

**Audience:**

Yes

**Broader Impact Concerns:**

This paper does not have Broader Impact Statement section, but considering the research topic the paper studies, I think it is not necessarily required.

**Claims And Evidence:**

Yes

**Requested Changes:**

### Mandantory
- [MR1] In Table 1, please explain the reason or intuition why RND-MiniGrid-DoorKey-16x16-RLeXprole is better than RND-MiniGrid-DoorKey-8x8-RLeXprole (0.6 vx. 0.0). This is not consistent with other algorithms (ICM, RE3, RIDE) and not intuitive because MiniGrid-DoorKey-8x8 should be easier than MiniGrid-DoorKey-16x16.
- [MR2] Related to Figure 3, please include the comprehensive Table of environment to explain observation space (image or sensor vectors), action space (discrete, continuous), reward (dense/sparse, available or not), termination, and which intrisic reward can be used (e.g. In Figure 31, only 3 intrisic rewards are used in AntUMaze). If the impremented intrisic rewards are agnostic to the environment, please ignore the last.
- [MR3] Please include the comprehensive Table of intrisic rewards to explain which type of algorithms can be combined such as (1)discrete/continuous action space, (2) observation space, (3) on-policy v.s. off-policy, (4) model-free v.s. model-based, etc. Please adjust the axes appropriately. This should improve the convinience for the users.
- [MR4] Please explain why the combination of reward is smaller than $\binom{8}{2} = 28$ pairs in Figure 8. If there are some criteria or strategy for the combination, please include it. If there are some infeasible combination due to some technical difficulties, please consider adding summary table to describe the available combinations.

### Recommendation
- [RR1] It would be great to add one or two sentences to explain why RLeXplore outperforms Original in Table 1. I guess this is due to the implementation details explored in Section 5 & Figure 4, but please highlight such exprolations realizes better implementations.
- [RR2] Please consider including the guide to descibire how the user can add novel intrinsic reward modules into RLeXplore.

**Strengths And Weaknesses:**

### Strengths
- [S1] RLeXplore provides a unified and modular framework that integrates various state-of-the-art intrinsic reward algorithms. Also, RLeXplore is an open-source library that provides high-quality implementations. This addresses the existing gap in standardized implementations, enhancing reproducibility and facilitating fair comparisons across different methods.
- [S2] RLeXplore can be seamlessly integrated with existing RL libraries such as Stable Baselines3, CleanRL, and RLLTE. This flexibility allows researchers to easily incorporate RLeXplore into their existing workflows.
- [S3] The paper presents a systematic study of implementation details that significantly impact the performance of intrinsic reward methods. By investigating factors such as observation normalization, reward normalization, update frequency, and weight initialization, the authors offer valuable insights into the practical aspects of intrinsically-motivated RL.
- [S4] The framework demonstrates the ability to reproduce and improve upon previously reported results across multiple environments, including SuperMarioBros, MiniGrid, and Procgen. This highlights the effectiveness of the RLeXplore implementations and the importance of the identified implementation details.

### Weaknesses
- Some actionable weaknesses are written in **Requested Changes**.
- [W1] As author mentioned, more complex algorithms like BYOL-Explore or RECODE were not included due to their computational demands and limited open-source availability. However, those algorithms can achieve SoTA performance ouperforming the algorithms impremented in this library. This poses a question if RLeXplore may only have a limited contribution to the progress of research on intrisic reward because the SoTA is not included.

---

> ### Author Response · Authors · 2025-03-13
> **Response to Reviewer K5wK (Part 1)**
>
> Dear Reviewer,
>
> We sincerely thank you for your insightful and constructive feedback, which has significantly improved the quality and clarity of our manuscript. For easy comparison, in the revision, we use the **\textcolor{magenta}{}** to mark the changes and the **\sout{}** command to indicate the parts that need to be deleted.
>
> **[MR1] In Table 1, please explain the reason or intuition why RND-MiniGrid-DoorKey-16x16-RLeXprole is better than RND-MiniGrid-DoorKey-8x8-RLeXprole (0.6 vx. 0.0). This is not consistent with other algorithms (ICM, RE3, RIDE) and not intuitive because MiniGrid-DoorKey-8x8 should be easier than MiniGrid-DoorKey-16x16.**
>
> We have checked our experiment settings and results. The issue is likely because we were comparing these results with a different number of training steps, 1M for DoorKey-8x8 and 10M for DoorKey-16x16, we’ve added the specific number of training steps in Table 1. Additionally, to validate the number of steps is the problem causing the discrepancy, we rerun the DoorKey-8x8 with the best hyperparameters gathered from the DoorKey-16x16 experiments, now its score is 0.82 (see Figure 9), which is better than the performance achieved in DoorKey-16x16.
>
> **[MR2] Related to Figure 2, please include the comprehensive Table of environment to explain observation space (image or sensor vectors), action space (discrete, continuous), reward (dense/sparse, available or not), termination, and which intrinsic reward can be used (e.g. In Figure 31, only three intrinsic rewards are used in AntUMaze). If the implemented intrinsic rewards are agnostic to the environment, please ignore the last.**
>
> Thanks for this valuable suggestion. We have added a detailed Table in Appendix B.1, which provides the detailed information of all the selected environments in our experiments. In addition, all the intrinsic rewards implemented in RLeXplore can operate in both discrete and continuous control tasks.
>
> **[MR3] Please include the comprehensive Table of intrinsic rewards to explain which type of algorithms can be combined such as (1)discrete/continuous action space, (2) observation space, (3) on-policy v.s. off-policy, (4) model-free v.s. model-based, etc. Please adjust the axes appropriately. This should improve the convenience for the users.**
>
> We have added a Table to show the API compatibility in **Section C.1**, which covers all the required attributes. All the intrinsic rewards implemented in RLeXplore can operate in both discrete and continuous control tasks and support both image-based and state-based observations.

---

> ### Author Response · Authors · 2025-03-13
> **Response to Reviewer K5wK (Part 2)**
>
> **[MR4] Please explain why the combination of reward is smaller than 28 pairs in Figure 8. If there are some criteria or strategy for the combination, please include it. If there are some infeasible combination due to some technical difficulties, please consider adding summary table to describe the available combinations.**
>
> Our study on mixed intrinsic rewards is inspired by [1], which demonstrates that combining global and episodic bonuses can provide more comprehensive exploration incentives and lead to significant performance gains in hard-exploration environments. In Figure 8, certain intrinsic reward combinations are not included for the following reasons:
>
> - As shown in Appendix 1, NGU is composed of an RND module and a PseudoCounts module, meaning it already integrates both global and episodic bonuses. Similarly, RIDE incorporates a PseudoCounts module to discount its final intrinsic reward. As for Disagreement, it is conceptually similar to the ICM. So we omitted these algorithms from explicit pairwise combinations to reduce redundancy and simplify the experiments.
>
> - Based on [1], we further investigate the potential of the “global+global”, as RND, RE3, and RIDE have distinct design philosophies. It was really interesting to study the “chemical” reaction of their mixture.
>
> - Why is there no “episodic+episodic”? First, the RE3 and the implemented PseduoCounts actually follow a similar insight, i.e., use KNN to estimate the density of the state visitation distribution and transform it into particle-based rewards. As a result, combining them would likely lead to redundant reward signals rather than a meaningful enhancement. Furthermore, episodic bonuses typically operate within limited landscapes, meaning they lack the diversity needed to effectively complement one another. In contrast, combining global and episodic rewards leverages different exploration incentives, leading to more robust performance improvements.  So we omit it here.
>
> - RLeXplore provides a .Fabric() that allows developers to compose an arbitrary number of intrinsic rewards regardless of their design philosophies. This modular framework enables researchers to experiment with various reward compositions beyond the ones we evaluated in this study. Future work could leverage this capability to explore a broader range of intrinsic reward combinations and assess their effectiveness in different exploration settings.
>
> [1] Henaff M, Jiang M, Raileanu R. A study of global and episodic bonuses for exploration in contextual mdps[C]//International Conference on Machine Learning. PMLR, 2023: 12972-12999.
>
>
> **[RR1] It would be great to add one or two sentences to explain why RLeXplore outperforms Original in Table 1. I guess this is due to the implementation details explored in Section 5 & Figure 4, but please highlight such exprolations realizes better implementations.**
>
> The performance gains are primarily a result of improvements in our implementation and careful analysis of the best design decisions shown in Figure 4, as you note.
>
> As mentioned in the last paragraph in the Introduction:
>
> “Our results highlight the importance of thoughtful implementation design for intrinsic rewards, showing that naive implementations can lead to suboptimal performance. Through carefully studied design decisions, we demonstrate significant performance gains.”
>
> However, as noted in the paper, a portion of the gains may also stem from using newer RL algorithms to optimize the intrinsic rewards. For instance, although we mainly use PPO with fixed hyperparameters across most objectives, some objectives were originally optimized using other algorithms like IMPALA or A3C. We provide a detailed discussion of these factors in Appendix D.6. We note that the evaluation environments remain the same as in the original works from which we report the results in Table 1.

---

> ### Author Response · Authors · 2025-03-13
> **Response to Reviewer K5wK (Part 3)**
>
> **[RR2] Please consider including the guide to describe how the user can add novel intrinsic reward modules into RLeXplore.
> We have now included a guide in the revised manuscript that describes how users can add novel intrinsic reward modules into RLeXplore.**
>
> As presented in Figure 1, all intrinsic rewards are abstracted from a base reward class that requires the implementation of two key functions:
>
> - `.compute()`: This function takes a batch of on-policy trajectories to compute the intrinsic rewards. It is automatically called before the PPO update.
>
> - `.update()`: This function uses the same batch of on-policy trajectories to update the modules used for computing the intrinsic rewards.
>
> To implement a new intrinsic reward, users simply need to inherit from this base reward class and define these two methods in a new script. Additionally, many network modules (such as the Atari CNN or ResNet CNN) can be imported directly, allowing users to leverage existing components as references for their own implementations.
>
> We have added Appendix E to present this information:
>
> “Adding New Intrinsic Reward Modules: In RLeXplore, all intrinsic reward methods inherit from a base reward class that requires two functions to be implemented: compute() and update(). The compute() function processes a batch of on-policy trajectories to calculate intrinsic rewards and is automatically called prior to the PPO update, while the update() function uses the same trajectories to update the associated modules. To integrate a new intrinsic reward method, users only need to create a new script that inherits from the base reward class and implements these two functions. Moreover, many pre-defined network modules (e.g., Atari CNN, ResNet CNN) are readily available for import, allowing users to use the currently implemented intrinsic rewards as templates for their own implementations”
>
>
> Hope that these can help address the raised comments. If you have any further comments, please don't hesitate to let us know. Thanks!

---

### Review · Reviewer_Lm7H · 2025-03-03

**Summary Of Contributions:**

The authors introduce a software framework for evaluating intrinsic motivation methods in reinforcement learning. They accompany their framework with extensive evaluations of the implemented methods across different tasks, particularly focusing on the often overlooked importance of hyper parameters.

**Audience:**

Yes

**Claims And Evidence:**

Yes

**Requested Changes:**

* I would like to see a performance comparison of Disagreement, NGU, and PseudoCounts to the results reported in the corresponding papers in Appendix D.
* After reading Section 5.1, I was not sure whether the "Combined" experiments were using the best hyper parameters across methods or per method. It would be hence worth highlighting whether the authors could identify one universally well working set of hyper parameters for all their implemented methods or not.
* Figure 5 could benefit from highlighting which investigations belong to which research questions (like in Figure 4).
* I did not understand why the authors outline URL algorithms in Section 2.4. As far as I understand, the authors did not implement or evaluate any URL methods. Mentioning the URL benchmark could be done without talking about URL methods. Therefore, I would recommend to not make this discussion a sub-section in order to not confuse readers about the contents of the framework.
* I would recommend the authors to add references to Appendix B, F, and G from the main text.

**Strengths And Weaknesses:**

Strengths:
* The paper is well-written and easy to follow
* The investigation of hyper parameters is interesting, well presented, and most importantly important for researchers and practitioners.
* The described API of the framework looks clean and easy to use

The only shortcoming that I see within the scope of the paper (a framework for evaluating intrinsic reward methods) is that it lacks a clear comparison between the performance of the re-implementations and the original results for Disagreement, NGU, and PseudoCounts.

---

> ### Author Response · Authors · 2025-03-13
> **Response to Reviewer Lm7H**
>
> Dear Reviewer,
>
> We sincerely thank you for your insightful and constructive feedback, which has significantly improved the quality and clarity of our manuscript. For easy comparison, in the revision, we use the **\textcolor{magenta}{}** to mark the changes and the **\sout{}** command to indicate the parts that need to be deleted.
>
> **[RC1] I would like to see a performance comparison of Disagreement, NGU, and PseudoCounts to the results reported in the corresponding papers in Appendix D.**
>
> Table 15 now provides a direct performance comparison on the ALE-5 benchmark between our RLeXplore implementations and the original ones reported in the literature (detailed in Appendix D).
>
> - **NGU**: Since no public codebase or dataset is available for NGU, we extracted its baseline performance numbers directly from the paper. Despite operating under a more limited training budget, our NGU implementation still achieves competitive performance compared to the published results.
>
> - **Disagreement**: We used the official repository (https://github.com/pathak22/exploration-by-disagreement) and trained the model for 1M frames. The scores we obtained match the results reported in [1], confirming that our implementation faithfully reproduces the expected performance.
>
> - **PseudoCounts**: PseudoCounts is integrated as a component of the NGU method and does not have an independent implementation. Therefore, a separate baseline comparison is not available.
>
> Overall, the improved performance of RLeXplore’s implementations is largely attributable to the extra effort put into fine-tuning low-level details, which helps to optimize the interplay of the various components.
>
> [1] Bai C, Wang L, Han L, et al. Dynamic bottleneck for robust self-supervised exploration[J]. Advances in Neural Information Processing Systems, 2021, 34: 17007-17020.
>
>
> **[RC2] After reading Section 5.1, I was not sure whether the "Combined" experiments were using the best hyper parameters across methods or per method. It would be hence worth highlighting whether the authors could identify one universally well working set of hyper parameters for all their implemented methods or not.**
>
> The "Combined" experiments use the best hyperparameters selected per method. We have clarified this in the caption of Figure 5. While no single set of hyperparameters works universally well for all intrinsic rewards, some choices can significantly impact performance. For example, a low update proportion (e.g., UP = 1%) can dramatically reduce performance, whereas others, like Min-Max reward normalization (RN), can provide substantial improvements, as shown in Figure 5.
>
> **[RC3] Figure 5 could benefit from highlighting which investigations belong to which research questions (like in Figure 4).**
>
> Thanks for the suggestion! We’ve added the RQ indices to the y-axis tick labels. Please refer to the revised paper.
>
> **[RC4] I did not understand why the authors outline URL algorithms in Section 2.4. As far as I understand, the authors did not implement or evaluate any URL methods. Mentioning the URL benchmark could be done without talking about URL methods. Therefore, I would recommend to not make this discussion a sub-section in order to not confuse readers about the contents of the framework.**
>
> We have removed the section discussing URL algorithms from Section 2.4. In the revised version, we now only mention the URL benchmark for contextual reference without outlining any specific URL methods.
>
>
> **[RC5] I would recommend the authors to add references to Appendix B, F, and G from the main text.**
>
> Thanks for the suggestion! We’ve added the references in Section 4.1 (Appendix F, G) and Section 5 (Appendix B).
>
> Hope that these can help address the raised comments. If you have any further comments, please don't hesitate to let us know. Thanks!

---

### Decision · Action_Editor_LRZB · 2025-03-29

**Recommendation:** Accept with minor revision

**Comment:**

The reviewers are in consensus that the revision strongly improved the paper. K5wK: "They sufficiently addressed my concerns and comments with paper revision. The unified benchmark and codebase would promote future research on intrinsic-reward-based exploration in reinforcement learning" and dYVt "The authors have effectively addressed all concerns and open questions raised during the rebuttal phase. The revised manuscript is an improved version that incorporates the necessary clarifications and refinements in response to reviewers’ feedback."

In the final version of the paper the author should address two minor clarifications following:

* In response to reviewer K5wK, the authors performed an evaluation of the RND algorithm in the MiniGrid-DoorKey-8x8 environment with an increased number of steps, confirming the expected effect that under the same number of steps (10M) RND would perform better in MiniGrid-DoorKey-8x8 than in MiniGrid-DoorKey-16x16. However, in the updated paper, the authors list the newly achieved performance as if it has been achieved with 1M steps.

* The authors should add an explanation of the new Table 15 in Appendix D.5.

**Audience:**

The paper is of relevance to the reinforcement learning research community.

**Claims And Evidence:**

The paper presents a framework and benchmarking environments for the intrinsic rewards. The benchmark includes implementations of eight intrinsic reward algorithms, study of the implementations details, and evaluations across different tasks, highlighting hyper-parameters. The reviewers found the paper well-written, easy to ready, and very helpful for advancing reinforcement learning research.

The reviewers are in consensus that the revision strongly improved the paper. In the final version of the paper the author should address the following:

* In response to reviewer K5wK, the authors performed an evaluation of the RND algorithm in the MiniGrid-DoorKey-8x8 environment with an increased number of steps, confirming the expected effect that under the same number of steps (10M) RND would perform better in MiniGrid-DoorKey-8x8 than in MiniGrid-DoorKey-16x16. However, in the updated paper, the authors list the newly achieved performance as if it has been achieved with 1M steps.

* The authors should add an explanation of the new Table 15 in Appendix D.5.